# Inverse Virtual Try-On: Generating Multi-Category Product-Style Images from Clothed Individuals

*Davide Lobba[1,3], *Fulvio Sanguigni[2,3], †Bin Ren[1,3], Marcella Cornia[2], Rita Cucchiara[2], Nicu Sebe[1]
[1]University of Trento    [2]University of Modena and Reggio Emilia    [3]University of Pisa

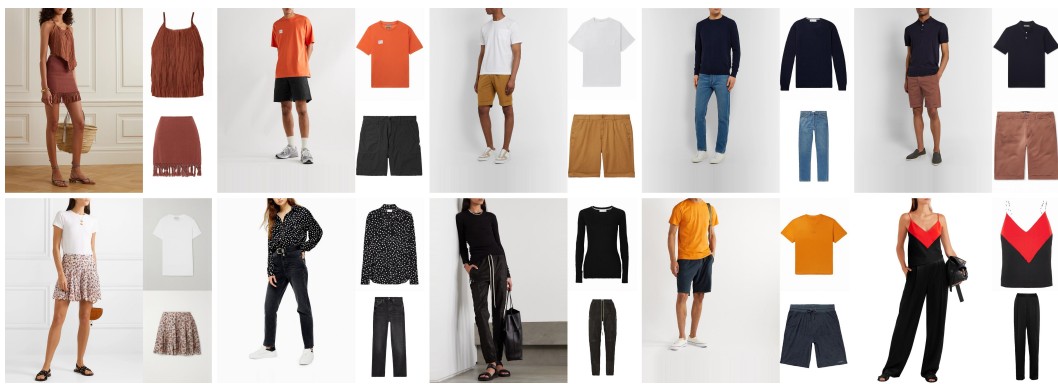

Figure 1: Visual results produced by our proposed text-enhanced multi-category virtual try-off architecture, *i.e.*, TEMU-VTOFF. Given a clothed input person image, the proposed model reconstructs the clean, in-shop version of the worn garment. Our model handles various garment types and preserves both structural fidelity and fine-grained textures, even under occlusions and complex poses, thanks to its multimodal attention and garment-alignment design.

## Abstract

Virtual try-on (VTON) has been widely explored for rendering garments onto person images, while its inverse task, virtual try-off (VTOFF), remains largely overlooked. VTOFF aims to recover standardized product images of garments directly from photos of clothed individuals. This capability is of great practical importance for e-commerce platforms, large-scale dataset curation, and the training of foundation models. Unlike VTON, which must handle diverse poses and styles, VTOFF naturally benefits from a consistent output format in the form of flat garment images. However, existing methods face two major limitations: *(i)* exclusive reliance on visual cues from a single photo often leads to ambiguity, and *(ii)* generated images usually suffer from loss of fine details, limiting their real-world applicability. To address these challenges, we introduce **TEMU-VTOFF**, a **T**ext-**E**nhanced **MU**lti-category framework for **VTOFF**. Our architecture is built on a dual DiT-based backbone equipped with a multimodal attention mechanism that jointly exploits image, text, and mask information to resolve visual ambiguities and enable robust feature learning across garment categories. To explicitly mitigate detail degradation, we further design an alignment module that refines garment structures and textures, ensuring high-quality outputs. Extensive experiments on VITON-HD and Dress Code show that TEMU-VTOFF achieves new state-of-the-art performance, substantially improving both visual realism and consistency with target garments. Code and models are available at: `https://temu-vtoff-page.github.io/`.

## 1 Introduction

Unlike virtual try-on (VTON), whose goal is to dress a given clothing image on a target person image, in this paper, we focus exactly on the opposite, virtual try-off (VTOFF), whose purpose is

---

* indicates equal contribution, † indicates corresponding author.

to generate standardized product images from real-world clothed individual photos. Compared to VTON, which often struggles with the ambiguity and diversity of valid outputs, such as stylistic variations in how a garment is worn, VTOFF benefits from a clearer output objective: *reconstructing a consistent, lay-down-style image of the garment*. This reversed formulation facilitates a more objective evaluation of garment reconstruction quality.

The fashion industry, a trillion-dollar global market, is increasingly integrating AI and computer vision to optimize product workflows and enhance user experience. VTOFF, in this context, offers substantial value: it enables the automatic generation of tiled product views, which are essential for tasks such as image retrieval, outfit recommendation, and virtual shopping. However, acquiring such lay-down images is expensive and time-consuming for retailers. VTOFF provides a scalable alternative by leveraging images of garments worn by models or customers, transforming them into standardized catalog views through image-to-image translation techniques.

Despite the success of GANs (Goodfellow et al., 2014) and Latent Diffusion Models (LDMs) (Rombach et al., 2022) in image translation tasks (Siarohin et al., 2019; Ren et al., 2023b; Isola et al., 2017; Tumanyan et al., 2023), current VTOFF solutions face notable limitations. Existing models (Velioglu et al., 2024; Xarchakos & Koukopoulos, 2024) struggle to accurately reconstruct catalog images from dressed human inputs. This limitation arises from a fundamental architectural mismatch: these approaches repurpose VTON pipelines by merely reversing the input-output roles, without addressing the unique challenges of the VTOFF task. Moreover, the high visual variability of real-world images – due to garment wear category (*e.g.*, upper-body), pose changes, and occlusions – makes it difficult for these models to robustly extract garment features while preserving fine-grained patterns. On the opposite side, we design a dedicated architecture tailored for the VTOFF task.

Recent advances in diffusion models demonstrate that DiT-based architectures (Peebles & Xie, 2023), especially when combined with flow-matching objectives (Lipman et al., 2023), surpass traditional U-Net and DDPM-based approaches (Rombach et al., 2022). Inspired by these findings, we propose **TEMU-VTOFF**, a Text-Enhanced MUlti-category Virtual Try-OFF architecture based on a dual-DiT framework. Specifically, we exploit the representational strength of DiT in two distinct ways: *(i)* the first Transformer component focuses on extracting fine-grained garment features from complex, detail-rich person images; and *(ii)* the second DiT is specialized for generating the clean, in-shop version of the garment. To support this design, we further adapt the base DiT architecture to accommodate the task-specific input modalities. To further enhance alignment, we introduce an external garment aligner module and a novel supervision loss that leverages clean garment references as guidance, further improving quality of generated images.

Our contribution can be summarized as follows:

- **Multi-Category Try-Off.** We present a unified framework capable of handling multiple garment types (upper-body, lower-body, and full-body clothes) without requiring category-specific pipelines.

- **Multimodal Hybrid Attention.** We introduce a novel attention mechanism that integrates garment textual descriptions into the generative process by linking them with person-specific features. This helps the dual-DiT architecture synthesize the garments more accurately.

- **Garment Aligner Module.** We design a lightweight aligner that conditions generation on clean garment images, replacing conventional denoising objectives. This leads to better alignment consistency on the overall dataset and preserves more precise visual retention.

- Extensive experiments on the Dress Code and VITON-HD datasets demonstrate that TEMU-VTOFF outperforms prior methods in both the quality of generated images and alignment with the target garment, highlighting its strong generalization capabilities.

## 2 RELATED WORK

**Virtual Try-On.** As a core task in the fashion domain, VTON has been extensively studied by the computer vision and graphics communities due to its practical potential (Bai et al., 2022; Cui et al., 2021; Fele et al., 2022; Ren et al., 2023a). Existing methods are broadly categorized into warping-based (Chen et al., 2023; Xie et al., 2023; Yan et al., 2023) and warping-free approaches (Zhu et al., 2023; Morelli et al., 2023; Baldrati et al., 2023; Zeng et al., 2024; Chong et al., 2025), with a growing shift from GAN-based (Goodfellow et al., 2020) to diffusion-based frameworks (Ho et al.,

2020; Song et al., 2021). VITON (Han et al., 2018) and its variants (Wang et al., 2018; Choi et al., 2021; Kang et al., 2021) improve garment alignment and synthesis quality, but often produce artifacts due to imperfect warping. To mitigate this, warping-free methods leverage diffusion models to bypass explicit deformation (Zhu et al., 2023; Morelli et al., 2023; Xu et al., 2025; Choi et al., 2024), using modified cross- or self-attention to condition generation on garment features. However, these pre-trained encoders tend to lose fine-grained texture details, motivating methods like StableVITON (Kim et al., 2024) to add dedicated garment encoders and attention modules, at increased computational cost. Lately, DiT-based works (Jiang et al., 2025; Zhu et al., 2024) show the benefits of Transformer-based diffusion models for high-fidelity garment to person transfer. Finally, some models adopt more elaborate conditioning strategies. For instance, LOTS introduces a pair-former module for handling multiple inputs (Girella et al., 2025), while LEFFA learns a flow field from averaged cross-attention maps and employs learnable tokens to stabilize attention values (Zhou et al., 2025). While most works focus on generating dressed images from separate person and garment inputs, the inverse problem (*i.e.*, reconstructing clean garment representations from worn images) remains underexplored.

**Virtual Try-Off.** While VTON has been extensively studied for synthesizing images of a person wearing a target garment, the recently proposed VTOFF task shifts the focus toward garment-centric reconstruction, aiming to extract a clean, standardized image of a garment worn by a person. TryOffDiff (Velioglu et al., 2024) introduces this task by leveraging a diffusion-based model with SigLIP (Zhai et al., 2023) conditioning to recover high-fidelity garment images. Building on this direction, TryOffAnyone (Xarchakos & Koukopoulos, 2024) addresses the generation of tiled garment images from dressed photos for applications like outfit composition and retrieval. By integrating garment-specific masks and simplifying the Stable Diffusion pipeline through selective Transformer tuning, it balances quality and efficiency. In both cases, these works have been designed for single-category scenarios, limiting scalability to diverse data collections. Recent efforts have started to address these limitations. MGT (Velioglu et al., 2025) extends VTOFF to multi-category scenarios by incorporating class-specific embeddings to handle diverse clothing types within a unified model. More ambitious efforts unify VTON and VTOFF in a single framework: Voost (Lee & Kwak, 2025) proposes a single DiT to learn both tasks, while One Model For All (Liu et al., 2025) introduces a partial diffusion mechanism to achieve a similar goal. On a different line, Any2AnyTryon (Guo et al., 2025) adapts FLUX (Labs, 2024) using a LoRA-based module (Hu et al., 2022) for this task. While these works reflect a shift from person-centric synthesis to garment-centric understanding, they still suffer from structural artifacts (*e.g.*, shape, neckline, waist) and inaccurate colors or textures. We hypothesize that this mismatch is due to a too generic architectural choice, not tailored for the specific needs of the VTOFF setting. In this work, we focus on existing VTOFF open problems, such as multi-category adaptation, occlusions, and complex human poses, and propose a novel VTOFF-specific architecture enhanced with text and fine-grained mask conditioning and optimized with a garment aligner component that can improve the quality of generated garments.

**Conditioning Methods in Diffusion Models.** To overcome the limitations of text-only conditioning, many schemes leverage additional visual inputs such as segmentation maps, bounding boxes, poses, and points (Sun et al., 2024; Li et al., 2023; Chen et al., 2024; Nie et al., 2024; Lin et al., 2024; Wang et al., 2024). Prominent methods like ControlNet (Zhang et al., 2023) and T2I-Adapter (Mou et al., 2024) inject spatial conditions via auxiliary networks, while IP-Adapter (Ye et al., 2023) uses separate attention branches more suited to U-Nets than DiTs. Other works focus on unifying multiple conditions, either through modular controllers like Uni-ControlNet (Zhao et al., 2023) and dedicated adapters (Lin et al., 2025), or by concatenating visual embeddings directly into the Transformer input sequence (Tan et al., 2025; Wang et al., 2025; Xiao et al., 2025). Although these methods are effective for general personalization tasks (*i.e.*, placing an object from one image into another), they lack the fine-grained conditioning mechanism necessary to extract specific garment data from person images, a gap we address to unlock the VTOFF task.

## 3 METHODOLOGY

**Preliminaries.** The latest diffusion models are a family of generative architectures that corrupt a ground-truth image $z_0$ according to a flow-matching schedule (Lipman et al., 2023) defined as

$$z_t = (1-t)z_0 + t\epsilon_t \quad \epsilon \sim \mathcal{N}(0,1), \quad t \in [0,1].$$

(1)

Then, a diffusion model estimates back the injected noise $\epsilon_t$ through a Diffusion Transformer (DiT) (Peebles & Xie, 2023), obtaining a prediction $\hat{z}_0$. In Stable Diffusion 3 (SD3) (Esser et al.,

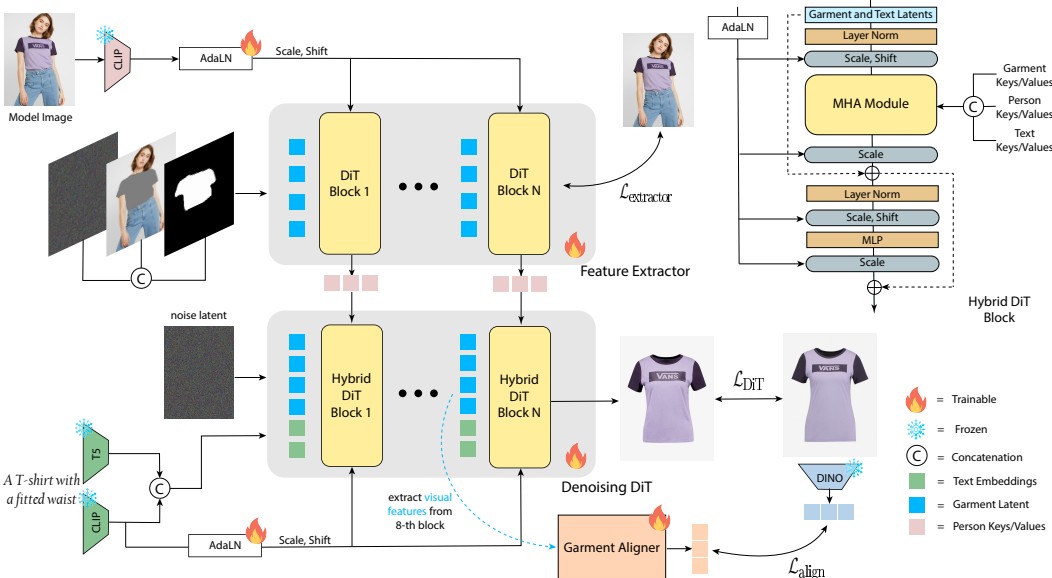

Figure 2: Overview of our method. The feature extractor $F_E$ processes spatial inputs (noise, masked image, binary mask) and global inputs (model image via AdaLN). The intermediate keys and values $\boldsymbol{K}^l_{\text{extractor}}$, $\boldsymbol{V}^l_{\text{extractor}}$ are injected into the corresponding hybrid blocks of the garment generator $F_D$. The main DiT then produces the final garment using the proposed MHA module. Training combines a diffusion loss on noise prediction with an alignment loss with DINOv2 features of the target garment.

2024), the 16-channel latent $\boldsymbol{z}_t \in \mathbb{R}^{\frac{H}{8} \times \frac{W}{8} \times 16}$ is obtained projecting the original RGB image $\boldsymbol{x} \in \mathbb{R}^{H \times W \times 3}$ with a variational autoencoder $\mathcal{E}$ (Kingma & Welling, 2013), obtaining $\boldsymbol{z} = \mathcal{E}(\boldsymbol{x})$, with $H, W$ being height and width of the image, and $f = 8$ the spatial compression ratio of the autoencoder. Finally, the model is trained according to an MSE loss function $\mathcal{L}_{\text{diff}}$:

$$\mathcal{L}_{\text{diff}} = \mathbb{E}_{\boldsymbol{z}_0, \epsilon_t, t} \left[ \| \epsilon_t - \epsilon_\theta(\boldsymbol{z}_t, t) \|^2 \right]. \tag{2}$$

**Overview.** An overview of our method is shown in Fig. 2. The objective is to generate an in-shop version of the garment worn by the person. A critical design choice lies in processing the dressed person image so as to extract meaningful information for injection into the denoising process. To this end, we adopt a dual-DiT architecture, built upon SD3, with the two models assigned to complementary roles. Firstly, we design the first DiT as a feature extractor $F_E$ that encodes the model image $\boldsymbol{x}_{\text{model}}$ and outputs its intermediate layer features at timestep $t = 0$ and not from subsequent timesteps, as we are interested in extracting clean features from $F_E$. This block is trained with a diffusion loss to generate the person image. Once trained, this model outputs meaningful key and value features of the dressed person. Secondly, the main DiT generates the garment $\boldsymbol{x}_g$ leveraging the intermediate features from $F_E$ in a modified textual-enhanced attention module.

### 3.1 DiT FEATURE EXTRACTOR

At inference time, the only available input is the clothed person image $\boldsymbol{x}_{\text{model}} \in \mathbb{R}^{H \times W \times 3}$, from which we also extract the mask. To encode this information, we compute the visual projection

$$\boldsymbol{e}^v_{\text{pool}} = \text{CLIP}(\boldsymbol{x}_{\text{model}}) \in \mathbb{R}^{2048},$$

which is then used to modulate the latent $\boldsymbol{z}_t$ through the AdaLN-estimated scale $\gamma$ and shift $\beta$:

$$\boldsymbol{y}_t = \text{MLP}(t, \boldsymbol{e}^v_{\text{pool}}),$$
$$\boldsymbol{z}_t \leftarrow \gamma(\boldsymbol{y}_t)\boldsymbol{z}_t + \beta(\boldsymbol{y}_t). \tag{3}$$

Existing VTON approaches rely on two visual inputs: the target garment and the person. In our case, however, the model can rely only on person features, from which it is more complex to extract the garment features. This shortcoming makes the CLIP vector $\boldsymbol{e}^v_{\text{pool}}$ the real bottleneck in a unified architecture setting, as the CLIP projection alone is too coarse to properly encode this information.

To address this, we propose introducing a dedicated feature extractor $F_E$, allowing $F_D$ to concentrate exclusively on the garment generation task. The architecture of $F_E$ mirrors that of the main SD3 DiT module $F_D$, with the only difference being its input layer, which is adapted to handle additional visual inputs in the channel dimension. The inputs are a global input with the person image $\boldsymbol{x}_{model} \in \mathbb{R}^{H \times W \times 3}$ encoded as $\boldsymbol{e}_{\text{pool}}^v$ and leveraged by the modulation layers of $F_E$, and a local spatial input as the channel-wise concatenation $\boldsymbol{z}_t' = [\boldsymbol{z}_t, M, \boldsymbol{x}_M] \in \mathbb{R}^{h \times w \times 33}$ of the latent $\boldsymbol{z}_t$, the encoded latent of the masked person image $x_M = \mathcal{E}(x_{model} \odot M) \in \mathbb{R}^{h \times w \times 16}$, where $\mathcal{E}$ is the SD3 VAE encoder, $h = H/f$ and $w = W/f$ denote the spatial dimensions after downsampling by the factor $f = 8$, and the interpolated binary mask $M \in \mathbb{R}^{h \times w \times 1}$ encoded through the Transformer projector $\mathcal{P} : \mathbb{R}^{h \times w \times 33} \to \mathbb{R}^{S \times d}$, with $S$ as sequence length and $d$ as embedding dimension.

This design choice is central to our method, as each layer output of the feature extractor $F_E$ retains meaningful intermediate representations of both the person and the garment. Leveraging these features offers three key advantages: *(i)* instead of the collapsed CLIP representation, we obtain expanded features of dimension $S \times d$; *(ii)* the $L$ layers of $F_E$ capture information at multiple granularities, progressing from coarse to fine (Avrahami et al., 2025; Skorokhodov et al., 2025), so that each layer $l$ conveys a different level of detail about the same image; *(iii)* since $F_E$ shares the same architecture as $F_D$, the features extracted at layer $l$ from $F_E$ are naturally better aligned with those of $F_D$. Motivated by these considerations, we extract the keys $\boldsymbol{K}_{\text{extractor}}^l$ and values $\boldsymbol{V}_{\text{extractor}}^l$ from every layer $l$ of $F_E$.

## 3.2 Dual-DiT Text-Enhanced Garment Try-Off

Without loss of generality, we will omit the index $l$ when referring to $\boldsymbol{Q}^l$ $\boldsymbol{K}^l$ and $\boldsymbol{V}^l$ of $F_E$ and $F_D$, since the conditioning scheme is applied uniformly across all layers. Given the extracted features $\boldsymbol{K}_{\text{extractor}}$ and $\boldsymbol{V}_{\text{extractor}}$, we propose to modify the SD3 attention scheme to incorporate such information, leading to our Multimodal Hybrid Attention (MHA).

**Multimodal Hybrid Attention.** Our new module seamlessly mix text information, latent features of the denoising DiT, and intermediate features from $F_E$. Inspired by the key findings in SD3 (Esser et al., 2024), we concatenate the text features with the visual inputs along the sequence length dimension, thus obtaining:

$$\boldsymbol{Q} = [\boldsymbol{Q}_{\boldsymbol{z}_t}, \boldsymbol{Q}_{\text{text}}] \quad \boldsymbol{K} = [\boldsymbol{K}_{\boldsymbol{z}_t}, \boldsymbol{K}_{\text{extractor}}, \boldsymbol{K}_{\text{text}}] \quad \boldsymbol{V} = [\boldsymbol{V}_{\boldsymbol{z}_t}, \boldsymbol{V}_{\text{extractor}}, \boldsymbol{V}_{\text{text}}]. \tag{4}$$

This module allows the features $\boldsymbol{Q}_{\text{text}}$ to attend both the latent projection $\boldsymbol{K}_{\boldsymbol{z}_t}$ and the extractor features $\boldsymbol{K}_{\text{extractor}}$. The resulting attention matrix $\boldsymbol{A}_{\text{MHA}}$ captures three key interactions: (i) $\boldsymbol{A}_{\text{text} \leftrightarrow \boldsymbol{z}_t}$, preserving pre-trained alignment between language and latent image tokens, (ii) $\boldsymbol{A}_{\boldsymbol{z}_t \leftrightarrow \text{extractor}}$, facilitating transfer between the input garment and the person representation, and (iii) $\boldsymbol{A}_{\text{text} \leftrightarrow \text{extractor}}$, grounding the text in the structural features provided by the extractor.

Text embeddings are constructed via the concatenation of CLIP (Radford et al., 2021)[1] and T5 (Raffel et al., 2020) encoders applied to the input caption $c$ as follows:

$$\boldsymbol{e}_{\text{text}} = [\text{CLIP}(c), \text{T5}(c)], \quad \text{with } \boldsymbol{e}_{\text{text}} \in \mathbb{R}^{77 \times 4096}. \tag{5}$$

Now we pose a relevant question: is it possible to disambiguate the garment category from the mask alone? A mask input can improve multi-category handling by acting as a *hard* discriminator between two garments, in contrast to text, which acts as a *soft* discriminator since it does not directly indicate the pixels occupied by the target garment. Therefore, the mask can help to visually force the model to retain only upper- or lower-body information but it can not tell much about the appearance of a garment, because it is highly warped together with the person, resulting in visual artifacts. Textual information is critical, together with mask information, to extract the category information of the garment. To address this, we decide to use also the global conditioning scheme provided by AdaLN (Huang & Belongie, 2017) in SD3. As shown in previous works (Garibi et al., 2025), these layers can be successfully leveraged to adapt "appearance" or "style" information into existing Transformer-based architectures. For this reason, we extract a pooled textual representation $\boldsymbol{e}_{\text{pool}} \in \mathbb{R}^{2048}$ of CLIP textual features of the caption $c$ and inject them into the model through the modulation layers, following Eq. 3. The pooled vector $\boldsymbol{e}_{\text{pool}} \in \mathbb{R}^{2048}$ encapsulates a coarser

---

[1]Following SD3, we consider the combined embedding from CLIP ViT-L and Open-CLIP bigG/14.

representation than the full textual embeddings $e_{\text{text}} \in \mathbb{R}^{77 \times 4096}$, thus being suitable for high-level information conditioning.

**Training.** We employ a two-stage training procedure: we train the module $F_E$ alone, detached from the dual DiT $F_D$, according to the diffusion loss $L_{\text{diff}}$ defined as follows:

$$\mathcal{L}_{\text{extractor}} = \mathbb{E}_{\boldsymbol{z}_0, \epsilon_t, t} \left[ \left\| \epsilon_t - F_E(\boldsymbol{z}'_t, \boldsymbol{x}_{\text{model}}, t) \right\|^2 \right]. \tag{6}$$

Then, we train the dual DiT $F_D$ following a diffusion loss with multiple conditioning signals:

$$\mathcal{L}_{\text{DiT}} = \mathbb{E}_{\boldsymbol{z}_g, \epsilon_t, t} \left[ \left\| \boldsymbol{z}_g - F_D(\boldsymbol{z}_t, \boldsymbol{e}_{\text{pool}}, F_E(\boldsymbol{z}'_0, \boldsymbol{x}_{\text{model}}, 0), t) \right\|^2 \right], \tag{7}$$

where $z_g = \mathcal{E}(x_g)$ is the latent representation of the target garment encoded by the VAE, and with $F_E(\boldsymbol{z}'_0, \boldsymbol{x}_{\text{model}}, 0)$ being the list of keys and values extracted from $F_E$ at timestep $t = 0$. We extract this list from $F_E$ at $t = 0$ and re-use them in $F_D$ for all subsequent timesteps, as we want to use key/values from clean data.

## 3.3 GARMENT ALIGNER

While our model is effective at generating realistic and structurally coherent garments, we observe occasional failures in preserving high-frequency details such as fine-grained textures and logos. We hypothesize two primary contributing factors: (i) the diffusion loss $\mathcal{L}_{\text{diff}}$, defined in the noise space, optimizes over perturbed latents rather than directly over image-space reconstructions, limiting its sensitivity to fine-grained patterns; and (ii) the inherent generation dynamics of diffusion models, where errors introduced in early timesteps – typically encoding low-frequency content – can accumulate and degrade the fidelity of high-frequency details in later stages. To mitigate this, we draw inspiration from REPA (Yu et al., 2025), and propose to explicitly align the internal feature representation of our DiT with that of a pre-trained vision encoder. Specifically, we encourage patch-wise consistency between the eighth Transformer block of our main DiT model $F_D$ and the corresponding features extracted from DINOv2 (Oquab et al., 2023).

Formally, let $\boldsymbol{h}_{\text{DiT}} \in \mathbb{R}^{3072 \times d}$ denote the token sequence obtained from the eighth Transformer block of the DiT decoder $F_D$, corresponding to a $64 \times 48$ patch grid with embedding dimension $d$. Separately, let $\boldsymbol{h}_{\text{enc}} \in \mathbb{R}^{1024 \times d'}$ be the $32 \times 32$ token grid extracted from a frozen DINOv2 encoder, with embedding dimension $d'$ (where $d' \neq d$). To bridge this mismatch, we introduce a lightweight garment aligner module composed of a convolutional neural network $\phi_{\text{CNN}} : \mathbb{R}^{64 \times 48 \times d} \to \mathbb{R}^{32 \times 32 \times d'}$ which is used to downsample the spatial token grid while preserving local structure and to project the token embeddings into the DINOv2 feature space. The aligned tokens are defined as $\tilde{\boldsymbol{h}}_{\text{DiT}} = \phi_{\text{CNN}}(\boldsymbol{h}_{\text{DiT}}) \in \mathbb{R}^{1024 \times d'}$.

We then enforce feature-level consistency via a cosine similarity loss:

$$\mathcal{L}_{\text{align}} = -\mathbb{E}_{\boldsymbol{z}_g, \epsilon_t, t} \left[ \frac{1}{N} \sum_{i=1}^{N} \cos\left( \tilde{\boldsymbol{h}}_i^{\text{DiT}}, \boldsymbol{h}_i^{\text{enc}} \right) \right], \tag{8}$$

where $\tilde{\boldsymbol{h}}_i^{\text{DiT}}$ and $\boldsymbol{h}_i^{\text{enc}}$ are the $i$-th aligned and reference tokens, respectively, $i$ is the patch index, $N$ is the total number of tokens, and $\cos$ is the cosine similarity. It is important to note that the garment aligner is strictly a training-time component used to compute $\mathcal{L}_{align}$. It is discarded during inference, adding no computational overhead to the generation process.

**Overall Loss Function.** The garment aligner is applied in the second stage of our training. Our final training objective combines the standard diffusion loss $\mathcal{L}_{\text{DiT}}$ with the garment alignment loss $\mathcal{L}_{\text{align}}$ previously introduced. The overall objective is thus defined as:

$$\mathcal{L}_{\text{total}} = \mathcal{L}_{\text{DiT}} + \lambda \cdot \mathcal{L}_{\text{align}}, \tag{9}$$

where $\lambda$ is a hyperparameter that balances the contribution of the two loss components.

## 4 EXPERIMENTS

### 4.1 COMPARISON WITH THE STATE OF THE ART

We conduct our experiments using two publicly available fashion datasets: VITON-HD (Choi et al., 2021) and Dress Code (Morelli et al., 2022). VITON-HD contains only upper-body garments and

Table 1: Quantitative results on the Dress Code dataset, considering both the entire test set and the three category-specific subsets. ↑ indicates higher is better, ↓ lower is better.

| | **All** | | | | | | **Upper-Body** | | | | | |
|---|---|---|---|---|---|---|---|---|---|---|---|---|
| Method | SSIM ↑ | PSNR ↑ | LPIPS ↓ | DISTS ↓ | FID ↓ | KID ↓ | SSIM ↑ | PSNR ↑ | LPIPS ↓ | DISTS ↓ | FID ↓ | KID ↓ |
| TryOffDiff | - | - | - | - | - | - | 76.59 | 11.54 | 40.62 | 29.04 | 37.97 | 17.30 |
| Any2AnyTryon | 77.56 | 12.67 | 35.17 | 25.17 | 12.32 | 3.65 | 76.61 | 12.27 | 38.99 | 25.78 | 15.77 | 3.22 |
| MGT | **77.77** | 11.99 | 35.37 | 27.28 | 13.47 | 5.28 | **76.77** | 11.44 | 39.70 | 28.13 | 19.49 | 6.87 |
| **TEMU-VTOFF** | 75.95 | **12.90** | **31.46** | **18.66** | **5.74** | **0.65** | 74.54 | **12.51** | **35.48** | **19.75** | **10.94** | **0.76** |

| | **Lower-Body** | | | | | | **Dresses** | | | | | |
|---|---|---|---|---|---|---|---|---|---|---|---|---|
| Method | SSIM ↑ | PSNR ↑ | LPIPS ↓ | DISTS ↓ | FID ↓ | KID ↓ | SSIM ↑ | PSNR ↑ | LPIPS ↓ | DISTS ↓ | FID ↓ | KID ↓ |
| Any2AnyTryon | **78.15** | **12.42** | 34.72 | 25.87 | 30.06 | 12.01 | 77.93 | 13.32 | 31.80 | 23.86 | 19.20 | 6.27 |
| MGT | 77.29 | 11.64 | 36.31 | 28.00 | 25.98 | 9.64 | 79.26 | 13.09 | 30.11 | 25.70 | 19.09 | 5.74 |
| **TEMU-VTOFF** | 73.94 | 12.14 | **34.60** | **19.57** | **13.83** | **2.04** | **79.39** | **14.36** | **24.32** | **16.67** | **11.29** | **0.59** |

Figure 3: Qualitative comparison on the Dress Code dataset between images generated by TEMU-VTOFF and those generated by competitors.

represents a single-category setting, while Dress Code includes multiple categories (*i.e.*, dresses, upper-body, and lower-body garments) enabling evaluation of the generalization capabilities of our methods across diverse garment types. To evaluate the proposed TEMU-VTOFF architecture, we use a combination of perceptual, structural, and distributional similarity metrics. Specifically, as VTOFF is a paired generation setting, we mainly rely on reference-based like SSIM (Wang et al., 2004), PSNR (Wang et al., 2004), LPIPS (Zhang et al., 2018), and DISTS (Ding et al., 2020), alongside FID (Parmar et al., 2022) and KID (Bińkowski et al., 2018). We compare our approach against recent VTOFF methods, including TryOffDiff (Velioglu et al., 2024), TryOffAnyone (Xarchakos & Koukopoulos, 2024), MGT (Velioglu et al., 2025), Voost (Lee & Kwak, 2025), One Model For All (Liu et al., 2025), and Any2AnyTryon (Guo et al., 2025). For TryOffAnyone, Voost, and One Model For All, we report the results only on the VITON-HD dataset because they have not been trained on the Dress Code dataset. Additionally, we retrain TryOffDiff on Dress Code using the official code and hyperparameters provided by the authors. Since TryOffDiff is not designed to handle multi-category garments, we report results only for the upper-body category.

**Results on the Dress Code Dataset.** Table 1 reports the experimental results on the Dress Code dataset. As observed, our method outperforms existing state-of-the-art approaches across most evaluation metrics and garment categories. These results indicate that our approach is category-agnostic and benefits from the joint use of textual garment descriptions and fine-grained masks. Consequently, our model achieves a better perceptual quality and closer alignment with the ground-truth distribution compared to competing methods. Performance on lower-body garments is slightly lower due to the class imbalance in the Dress Code dataset, which contains significantly fewer lower-body samples (∼9k) compared to upper-body (∼15k) and full-body dresses (∼29k).

In Fig. 3, we provide qualitative results comparing TEMU-VTOFF with competitors. These examples highlight the challenges posed by the diverse set of categories in Dress Code. As shown, MGT and Any2AnyTryon frequently struggle to preserve key visual attributes such as color, texture, and shape. In contrast, our method is able to closely match the target garment across all categories, demonstrating a clear improvement in generation quality.

**Results on the VITON-HD Dataset.** In Table 2, we report the quantitative results on VITON-HD. In this setting, TEMU-VTOFF sets a new state-of-the-art across the majority of metrics, achieving the

Table 2: Quantitative results on the VITON-HD dataset. ↑ indicates higher is better, ↓ lower is better. † denotes results taken directly from the original papers.

| Method | SSIM ↑ | PSNR ↑ | LPIPS ↓ | DISTS ↓ | FID ↓ | KID ↓ |
|---|---|---|---|---|---|---|
| TryOffDiff | 75.53 | 11.65 | 39.56 | 25.53 | 17.49 | 5.30 |
| TryOffAnyone | 75.90 | 12.00 | 35.26 | 23.47 | 12.74 | 2.85 |
| Any2AnyTryon | 75.72 | 12.00 | 37.95 | 24.32 | 12.88 | 3.01 |
| MGT† | **78.10** | - | 36.30 | 24.70 | 21.90 | 8.90 |
| Voost† | - | - | - | - | 10.06 | 2.48 |
| One Model for All† | - | - | **22.50** | 19.20 | 9.12 | 1.49 |
| **TEMU-VTOFF** | 77.21 | **13.38** | 28.44 | **18.04** | **8.71** | **1.11** |

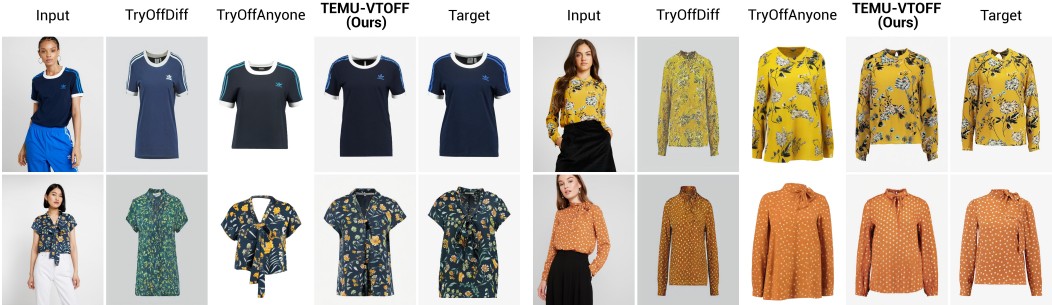

Figure 4: Qualitative comparison on the VITON-HD dataset between images generated by TEMU-VTOFF and those generated by competitors.

best scores for DISTS, FID, and KID. This indicates a superior ability to reconstruct structural details and to match the distribution of the ground-truth images. Notably, One Model for All achieves a competitive LPIPS score, which we cite from the original paper as public checkpoints are unavailable. Our method, however, achieves more robust performance on FID, KID, and DISTS. Since LPIPS and DISTS are critical in paired settings, we provide a qualitative analysis (Sec. E of the Appendix) showing LPIPS can diverge from human judgment, while DISTS aligns more reliably.

Overall, our method achieves solid improvements on VITON-HD, although the performance gains are less pronounced than on Dress Code. This is expected, as VITON-HD focuses exclusively on upper-body garments and is therefore a simpler benchmark. In contrast, the diverse and multi-category nature of Dress Code, with dresses, skirts, and pants, highlights the advantages of our approach, where the joint use of textual descriptions and fine-grained masks proves critical for accurate garment reconstruction. Accordingly, the strengths of our method are most evident in complex, multi-category scenarios. A visual comparison on sample VITON-HD images is shown in Fig. 4, which further demonstrates the improved garment reconstruction quality of our proposed method.

## 4.2 ABLATION STUDIES

To assess the contribution of each component in our pipeline, we conduct a detailed ablation study on the Dress Code dataset reported in Table 3. We first investigate the impact of our dual-stream DiT architecture by removing the feature extractor $F_E$. In this setting, the garment aligner component is not employed. As shown, without the feature extractor, we experience a clear performance drop. In contrast, injecting $t = 0$ keys and values from $F_E$ into the generator component through the proposed MHA operator enables richer, multi-scale conditioning, leading to better results. Then, we analyze the impact of employing the garment aligner module. As it can be seen, the aligner module helps to improve perceptual fidelity, particularly in categories with complex structures such as dresses, confirming that the designed components plays a critical role to the final performance.

Finally, removing garment descriptions or fine-grained masks consistently reduces performance, with the largest drop when both are absent, confirming that masks act as spatial anchors while text provides complementary semantic and category-level cues. The best results are obtained when both inputs are present, highlighting their complementarity.

Table 3: Ablation study of the proposed components on the Dress Code dataset.

| | All | | | | | | Upper-body | | Lower-body | | Dresses | |
|---|---|---|---|---|---|---|---|---|---|---|---|---|
| | SSIM ↑ | PSNR ↑ | LPIPS ↓ | DISTS ↓ | FID ↓ | KID ↓ | DISTS ↓ | FID ↓ | DISTS ↓ | FID ↓ | DISTS ↓ | FID ↓ |
| *Effect of Dual-Stream DiT (w/o Garment Aligner)* | | | | | | | | | | | | |
| w/o feature extractor $F_E$ | 72.79 | 11.45 | 38.61 | 23.56 | 9.11 | 1.70 | 24.97 | 14.13 | 23.20 | 19.54 | 22.52 | 16.82 |
| **TEMU-VTOFF** | **76.01** | **12.85** | **30.84** | **20.63** | **5.91** | **0.78** | **21.77** | **11.26** | **22.26** | **14.22** | **17.86** | **11.86** |
| *Effect of Garment Aligner Component* | | | | | | | | | | | | |
| w/o garment aligner | **76.01** | 12.85 | **30.84** | 20.63 | 5.91 | 0.78 | 21.77 | 11.26 | 22.26 | 14.22 | 17.86 | 11.86 |
| **TEMU-VTOFF** | 75.95 | **12.90** | 31.46 | **18.66** | **5.74** | **0.65** | **19.75** | **10.94** | **19.57** | **13.83** | **16.67** | **11.29** |
| *Effect of Text and Mask Conditioning* | | | | | | | | | | | | |
| w/o text and masks | 71.04 | 10.92 | 39.68 | 25.20 | 9.63 | 3.17 | 23.71 | 19.75 | 65.85 | 49.19 | 20.12 | 15.47 |
| w/o text modulation | 73.88 | 12.28 | 34.63 | 22.54 | 7.75 | 1.52 | 24.02 | 13.48 | 24.33 | 18.13 | 19.27 | 13.30 |
| w/o fine-grained masks | 74.65 | 12.30 | 32.33 | 20.87 | 6.58 | 1.03 | 20.85 | 11.31 | 22.34 | 15.74 | 19.42 | 13.62 |
| **TEMU-VTOFF** | **75.95** | **12.90** | **31.46** | **18.66** | **5.74** | **0.65** | **19.75** | **10.94** | **19.57** | **13.83** | **16.67** | **11.29** |

(a) Evaluation of mask and text joint impact.  (b) Evaluation of garment aligner impact.

Figure 5: Qualitative comparisons demonstrating the contribution of each proposed component. Left: effect of text and mask conditioning as "soft" and "hard" proxies for multi-category modeling. Right: impact of the lightweight garment aligner, which guides training with clean garment supervision.

To better understand the strength of each component proposed in our approach, we provide a visual comparison on Dress Code in Fig. 5. When our method relies exclusively on visual features from the person, without any textual guidance, it can struggle to resolve ambiguities in the garment design, leading to errors in structural elements such as neckline, sleeve length, or overall fit. The introduction of a textual description provides essential structural cues, enabling the model to capture the intended garment type and style. The fine-grained mask then imposes a precise spatial boundary, enforcing a clean silhouette and sharp edges, which improves the overall shape and contour of the garment. Finally, the garment aligner further improves the visual fidelity by encouraging the reconstruction of high-frequency details. This results in improved textures and more accurate patterns, ensuring that the final generated garment is not only structurally correct but also rich in fine-grained detail.

## 4.3 CROSS-DATASET GENERALIZATION

To evaluate the robustness of TEMU-VTOFF against domain shifts and its ability to generalize to unseen garment types and poses, we conduct cross-dataset experiments. Specifically, we train our model on one dataset and evaluate it directly on the test set of the other dataset. We compare against MGT (Velioglu et al., 2025), TryOffDiff (Velioglu et al., 2024), and TryOffAnyone (Xarchakos & Koukopoulos, 2024). Note that we exclude Any2AnyTryon (Guo et al., 2025) from this specific analysis, as it is trained on a mixture of datasets including both Dress Code and VITON-HD, making the cross-dataset evaluation unfair. In Table 4, we present cross-dataset generalization results under two transfer settings. When trained on Dress Code and evaluated on VITON-HD, our model consistently surpasses MGT across both perceptual and distributional metrics, achieving a notably lower FID (20.39 vs. 23.11). Conversely, when trained on VITON-HD and tested on Dress Code (upper-body), our method again shows stronger generalization, obtaining an FID of 18.63 and clearly outperforming both TryOffDiff and TryOffAnyone.

Table 4: Quantitative comparison where models are trained on one dataset and tested on another to evaluate robustness to domain shift. ↑ indicates higher is better, ↓ lower is better.

| Method | SSIM ↑ | PSNR ↑ | LPIPS ↓ | DISTS ↓ | FID ↓ | KID ↓ |
|---|---|---|---|---|---|---|
| *Training on Dress Code → Test on VITON-HD* | | | | | | |
| MGT | **74.26** | 10.24 | 42.57 | 28.73 | 23.11 | 10.81 |
| **TEMU-VTOFF** | 72.80 | **10.85** | **40.19** | **24.20** | **20.39** | **7.00** |
| *Training on VITON-HD → Test on Dress Code (Upper-Body)* | | | | | | |
| TryOffDiff | **75.33** | 11.50 | 44.64 | 32.14 | 41.91 | 21.78 |
| TryOffAnyone | 71.96 | 10.52 | 47.14 | 27.54 | 24.45 | 9.84 |
| **TEMU-VTOFF** | 73.36 | **11.51** | **39.74** | **23.84** | **18.63** | **6.31** |

Table 5: VTON results from CatVTON trained on two Dress Code variants: the original dataset and the version augmented with TEMU-VTOFF-generated images.

| Training Dataset | All | | | | | | Upper-Body | | | | | |
| | SSIM ↑ | PSNR ↑ | LPIPS ↓ | DISTS ↓ | FID ↓ | KID ↓ | SSIM ↑ | PSNR ↑ | LPIPS ↓ | DISTS ↓ | FID ↓ | KID ↓ |
|---|---|---|---|---|---|---|---|---|---|---|---|---|
| Dress Code | **90.65** | 23.03 | 7.12 | 9.18 | 4.56 | 1.34 | **92.93** | 24.32 | **5.33** | 7.66 | 9.58 | 2.04 |
| **Dress Code (*Augm.*)** | **90.65** | **23.36** | **7.00** | **9.00** | **4.15** | **1.16** | 90.94 | **24.39** | **5.33** | **7.54** | **9.26** | **1.74** |

| Method | Lower-Body | | | | | | Dresses | | | | | |
| | SSIM ↑ | PSNR ↑ | LPIPS ↓ | DISTS ↓ | FID ↓ | KID ↓ | SSIM ↑ | PSNR ↑ | LPIPS ↓ | DISTS ↓ | FID ↓ | KID ↓ |
|---|---|---|---|---|---|---|---|---|---|---|---|---|
| Dress Code | 91.46 | 24.44 | 6.03 | 7.84 | 9.60 | 1.71 | **87.50** | 21.18 | 10.01 | 12.04 | 9.58 | 1.26 |
| **Dress Code (*Augm.*)** | **91.48** | **24.50** | **5.95** | **7.63** | **9.02** | **1.38** | **87.50** | **21.21** | **10.00** | **12.02** | **9.45** | **1.12** |

## 4.4 DOWNSTREAM UTILITY

To demonstrate the practical utility of TEMU-VTOFF, we evaluate its effectiveness as a data augmentation tool for the VTON downstream task. High-quality paired data (*i.e.*, person and in-shop garment) is expensive to acquire; our method addresses this by synthetically generating the "in-shop" garment directly from images of clothed people.

**Experimental Setup.** We use the Dress Code dataset (Morelli et al., 2022) and employ TEMU-VTOFF to generate synthetic in-shop garment images for the training samples. Specifically, for each person image in the upper- and lower-body categories, we generate the missing in-shop garment: the lower-body item for upper-body images and the upper-body item for lower-body images. This procedure augments the dataset with additional person-garment pairs generated by TEMU-VTOFF. We then employ CatVTON (Chong et al., 2025) (utilizing the SD 3 medium backbone) in two distinct settings: trained only on the standard Dress Code training set and trained on the Dress Code training set augmented with the synthetic pairs generated by TEMU-VTOFF.

**Results.** We evaluate the trained models on the official Dress Code test set. As shown in Table 5, the model trained with our augmented data achieves consistent improvements across both perceptual and distributional metrics. Notably, in the upper-body and lower-body categories, the augmented training yields lower FID scores (9.27 vs. 9.58 and 9.02 vs. 9.60, respectively). This confirms that the garments generated by TEMU-VTOFF preserve sufficient structural fidelity and texture details to serve as effective training signals, improving the generalization of state-of-the-art VTON models.

## 5 CONCLUSION

We presented TEMU-VTOFF, a novel architecture that pushes the boundaries of VTOFF for complex, multi-category scenarios. While existing methods often struggle with detail preservation and accurate reconstruction across diverse garment types, our approach is specifically designed to overcome these limitations. We achieve this through a novel dual-DiT framework that leverages multimodal hybrid attention to effectively fuse information from the person, the garment, and textual descriptions. To enhance realism, our proposed garment aligner module refines fine-grained textures and structural details. The effectiveness of our method is validated by state-of-the-art performance on standard VTOFF benchmarks, demonstrating its robustness in generating high-fidelity, catalog-style images.

**Acknowledgments.** This work was supported by EU Horizon projects ELIAS (No. 101120237) and ELLIOT (No. 101214398), and by the FIS project GUIDANCE (No. FIS2023-03251).

## ETHICS STATEMENT

Our method addresses the VTOFF task by generating flat, in-shop garment images from photos of dressed individuals. This enables a novel form of data augmentation in the fashion domain, allowing clean garment representations to be synthesized without manual segmentation or dedicated photoshoots. By bridging the gap between worn and catalog-like appearances, our approach can improve scalability for fashion datasets and support downstream applications such as retrieval, recommendation, and virtual try-on. However, as with any generative technology, there are important ethical and legal considerations. In particular, our model could be used to reconstruct garments originally designed by third parties, potentially raising issues of copyright and intellectual property infringement. We emphasize that our framework is intended for research and responsible use, and any deployment in commercial settings should ensure compliance with applicable copyright laws and respect for designer rights.

## REPRODUCIBILITY STATEMENT

This work uses only public datasets and open-source models for its training and evaluations. In the Appendix, we include all the implementation and dataset details to reproduce our results. In addition, we will publicly release the source code and trained models to further support reproducibility.

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

# A  ADDITIONAL DETAILS

## A.1  DATASETS DETAILS

**Dress Code.** In our experiments, we adopt the Dress Code dataset (Morelli et al., 2022), the largest publicly available benchmark for image-based virtual try-on. Unlike previous datasets limited to upper-body clothing, Dress Code includes three macro-categories: upper-body clothes with 15,363 pairs (*e.g.*, tops, t-shirts, shirts, sweatshirts), lower-body clothing with 8,951 pairs (*e.g.*, trousers, skirts, shorts), and full-body dresses with 29,478 pairs. The total number of paired samples is $53,792$, split into $48,392$ training images and $5,400$ test images at a resolution of $1024 \times 768$.

**VITON-HD.** Following previous literature, we also adopt VITON-HD (Choi et al., 2021), a publicly available dataset widely used in virtual try-on research. It is composed exclusively of upper-body garments and provides high-resolution images at $1024 \times 768$ pixels. The dataset contains a total of $27,358$ images, structured into $13,679$ garment-model pairs. These are split into $11,647$ training pairs and $2,032$ test pairs, each comprising a front-view image of a garment and the corresponding image of a model wearing it.

## A.2  IMPLEMENTATION DETAILS

For both the feature extractor and the diffusion backbone, we adopt Stable Diffusion 3 medium (Esser et al., 2024). All models are trained on a single node equipped with 4 NVIDIA A100 GPUs (64GB each), using DeepSpeed ZeRO-2 (Rajbhandari et al., 2021) for efficient distributed training. We use a total batch size of 32 and train each model for 30k steps, corresponding to approximately 960k images. Optimization is performed with AdamW (Loshchilov & Hutter, 2019), using a learning rate of $1 \times 10^{-4}$, a warmup phase of 3k steps, and a cosine annealing schedule. We train separate models per dataset to account for differences in distribution and garment structure. In all experiments, we set the alignment loss weight $\lambda$ equal to $0.5$.

We evaluate our method both with distribution-based metrics and per-sample similarity metrics. For the first group, we adopt FID (Parmar et al., 2022) and KID (Bińkowski et al., 2018) implementations derived from `clean-fid` PyTorch package[2]. Concerning the second group, we adopt both SSIM (Wang et al., 2004), PSNR (Wang et al., 2004), and LPIPS (Zhang et al., 2018) as they are the standard metrics adopted in the field to measure structural and perceptual similarity between a pair of images. We reuse the corresponding Python packages provided by TorchMetrics[3]. Finally, we adopt DISTS (Ding et al., 2020) as an additional sample-based similarity metric, as it correlates better with human judgment, as shown in previous works (Fu et al., 2023). We stick to the corresponding Python package[4] to compute it for our experiments.

## A.3  CAPTION EXTRACTION DETAILS

We leverage Qwen2.5-VL (Bai et al., 2025) to generate textual descriptions of the garments, which serve as semantic conditioning for our Dual-DiT architecture.

To ensure no ground-truth information leaks into the testing process, we employ two different generation pipelines for training and testing:

- **Training:** We generate captions using the *ground-truth in-shop garment images*. This ensures the model learns precise semantic correlations between visual features and textual attributes during optimization (see Fig. 6).

- **Inference:** At test time, the ground-truth garment image is strictly unavailable. Instead, the caption is generated directly from the *input person image*. Qwen2.5-VL is prompted to analyze the person's clothing and extract the relevant structural attributes. This ensures our method is fully applicable to "in-the-wild" scenarios where only the person image is known (see Fig. 7).

We define a variable `category` $\in$ {`dress`, `upper body`, `lower body`}. To avoid leaking color or texture information (which should be handled by the visual feature extractor $F_E$) and to

---

[2]`https://pypi.org/project/clean-fid/`
[3]`https://pypi.org/project/torchmetrics/`
[4]`https://pypi.org/project/DISTS-pytorch/`

focus solely on structural guidance, we utilize a strict prompt template. The prompt explicitly forbids the generation of non-structural attributes:

```
visual_attributes = {
    "dresses": ["Cloth Type", "Waist", "Fit", "Hem", "Neckline", "Sleeve
        Length", "Cloth Length"],
    "upper_body": ["Cloth Type", "Waist", "Fit", "Hem", "Neckline", "
        Sleeve Length", "Cloth Length"],
    "lower_body": ["Cloth Type", "Waist", "Fit", "Cloth Length"]
}
```

**System:** You are Qwen, created by Alibaba Cloud. You are a helpful assistant.

**User:**

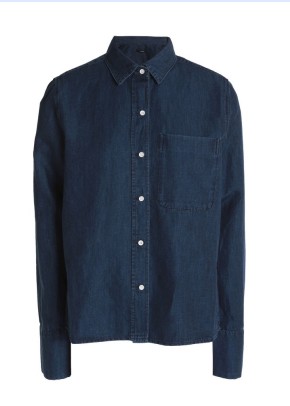

Use only visual attributes that are present in the image. Predict values of the following attributes: {`visual_attributes[category]`}. It's forbidden to generate the following visual attributes: colors, background, and textures/patterns. It's forbidden to generate unspecified predictions. It's forbidden to generate newline characters. Generate in this way: a `<cloth type> with <attributes description>`.

**Qwen Caption:** A denim shirt with a straight fit, long sleeves, and a button-down neckline. The hem is straight and the shirt appears to be of standard length.

Figure 6: Caption extraction pipeline (training stage).

We decide to generate structural-only attributes because our base model without text can already transfer colors and textures correctly from the person image to the generated garment image. The structural attributes are slightly different according to the three categories of clothing, as specified in `visual_attributes`. For example, the neckline can be specified for upper body and dresses (whole body garments), but not for lower body items.

## A.4 ALGORITHM

To provide a clear understanding of TEMU-VTOFF, we summarize the core components of our method in Algorithm 1. The pseudo-code outlines the sequential steps involved in training our dual-DiT architecture, including multimodal conditioning, the hybrid attention module, and the garment aligner component.

**System:** You are Qwen, created by Alibaba Cloud. You are a helpful assistant.

**User:**

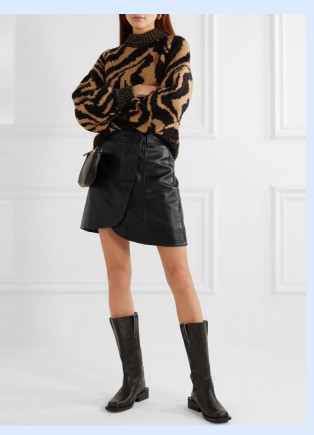

Use only visual attributes that are present in the image. Predict values of the following attributes: `{visual_attributes[category]}`. Do it only for the `category` garment. It's forbidden to generate the following visual attributes: colors, background, and textures/patterns. It's forbidden to generate unspecified predictions. It's forbidden to generate newline characters. Generate in this way: `a <cloth type> with <attributes description>`.

**Qwen Caption:** A leather skirt with a fitted waist and a short length.

Figure 7: Caption extraction pipeline (inference stage).

## B  ADDITIONAL QUANTITATIVE RESULTS AND ANALYSES

In this section, we report additional quantitative results and analyses on the effectiveness of the proposed components and design choices.

**Effect of Varying $\lambda$ Parameter.** We conducted an ablation study on the Dress Code dataset to assess the effect of the $\lambda$ regularization for the alignment of our main diffusion transformer $F_D$ and the DINOv2 features. We report the results in Table 6. As shown, $\lambda = 0.5$ is the overall best choice across all metrics.

**Effect of Varying the DiT Block $i$ Used for $\mathcal{L}_{align}$.** A critical design choice in our garment aligner module is determining which internal block of the DiT ($F_D$) should be aligned with the semantic features from DINOv2. To find the optimal depth, we conduct an ablation study on the Dress Code dataset, varying the block index $i \in \{6, 8, 12, 18\}$ within the 24-block SD 3 medium architecture.

The results are reported in the middle section of Table 6. We observe that aligning the 8th block yields the best overall performance. These results highlight a trade-off between structural guidance and generation flexibility, consistent with recent findings in representation alignment (Yu et al., 2025). The early-to-mid layers of the DiT capture the coarse semantic layout and structural essence of the image. Aligning these with DINOv2 ensures the generated garment respects the target structure while leaving subsequent layers free to refine details. As we move to deeper blocks, perceptual quality degrades. While these layers maintain high structural similarity, the distributional metrics worsen. This occurs because the deeper layers of a DiT are increasingly specialized in predicting the high-frequency noise (or flow velocity) required for the diffusion objective. Forcing these "noisy" layers to align with the "clean", invariant features of DINOv2 introduces an optimization conflict, ultimately smoothing out fine-grained textures and degrading realism.

**Analysis of Asynchronous Timestep Conditioning.** A critical design choice in our architecture is the use of a fixed timestep $t = 0$ for the feature extractor $F_E$, while the main denoising DiT

---

**Algorithm 1** TEMU-VTOFF: Dual-DiT and Garment Alignment for VTOFF

---

**Require:** Person image $\boldsymbol{x}_{\text{model}}$, garment caption $c$, binary mask $M$, target garment image $\boldsymbol{x}_g$
**Ensure:** Generated garment $\hat{\boldsymbol{x}}_g$

1: **Latent encoding:**
       Encode the target garment: $\boldsymbol{z}_g \leftarrow \mathcal{E}(\boldsymbol{x}_g)$
       Sample noise: $\epsilon_t \sim \mathcal{N}(0, 1)$
       Apply flow-matching: $\boldsymbol{z}_t \leftarrow (1 - t)\boldsymbol{z}_g + t \cdot \epsilon_t$
2: **Prepare masked spatial input:**
       Encode masked person image: $\boldsymbol{x}_M \leftarrow \mathcal{E}(\boldsymbol{x}_{\text{model}} \odot M)$
       Concatenate inputs: $\boldsymbol{z}'_t \leftarrow [\boldsymbol{z}_t, M, \boldsymbol{x}_M]$
3: **Extract modulation features:**
       $\boldsymbol{e}^v_{\text{pool}} \leftarrow \text{CLIP}(\boldsymbol{x}_{\text{model}})$
4: **Extract keys and values using feature extractor:**
       $\{\boldsymbol{K}^l_{\text{extractor}}, \boldsymbol{V}^l_{\text{extractor}}\}^N_{l=1} \leftarrow F_E(\boldsymbol{z}'_0, \boldsymbol{x}_{\text{model}}, t{=}0)$
5: **Encode text information:**
       Get pooled text embedding: $\boldsymbol{e}_{\text{pooled}} \leftarrow \text{CLIP}(c)$
       Get full sequence text features: $\boldsymbol{e}_{\text{text}} \leftarrow [\text{CLIP}(c), \text{T5}(c)]$
6: **Noise prediction:**
       $\hat{\epsilon}_t \leftarrow F_D(\boldsymbol{z}_t, \boldsymbol{e}_{\text{pooled}}, \boldsymbol{e}_{\text{text}}, \{\boldsymbol{K}^l_{\text{extractor}}, \boldsymbol{V}^l_{\text{extractor}}\}, t)$
       Compute diffusion loss: $\mathcal{L}_{\text{DiT}} \leftarrow \|\hat{\epsilon}_t - \epsilon_t\|^2$
7: **Align internal representations:**
       Extract DiT features: $\boldsymbol{h}_{\text{DiT}} \leftarrow$ tokens from 8th block of $F_D$
       Extract DINOv2 features: $\boldsymbol{h}_{\text{enc}} \leftarrow \text{DINOv2}(\boldsymbol{x}_g)$
       Align via projection: $\tilde{\boldsymbol{h}}_{\text{DiT}} \leftarrow \phi_{\text{CNN}}(\boldsymbol{h}_{\text{DiT}})$
       Compute alignment loss: $\mathcal{L}_{\text{align}} \leftarrow -\frac{1}{N}\sum_i \cos(\tilde{\boldsymbol{h}}^{\text{DiT}}_i, \boldsymbol{h}^{\text{enc}}_i)$
8: **Final objective:**
       Combine losses: $\mathcal{L}_{\text{total}} \leftarrow \mathcal{L}_{\text{DiT}} + \lambda \cdot \mathcal{L}_{\text{align}}$
9: **Decode final garment:**
       Run reverse process: $\hat{\boldsymbol{x}}_g \leftarrow \mathcal{D}(\hat{\boldsymbol{z}}_0)$

---

$F_D$ operates on a noisy latent $z_t$ at timestep $t > 0$. This raises an important question: could this discrepancy in timesteps lead to a misalignment in the feature space? In this section, we provide the rationale for this design choice, supported by concurrent work and a targeted ablation study.

Our primary motivation is to provide the main generator $F_D$ with the cleanest, most semantically rich conditioning signal possible. By extracting features from $F_E$ at $t = 0$ we ensure the conditioning information is completely free from stochastic noise inherent to the diffusion process. We hypothesize that injecting features from a noisy timestep $t > 0$ would introduce an additional, confounding source of noise into the generation process, thereby degrading the quality of the final output. The key to our method is that the MHA module is specifically trained to bridge this temporal gap; it learns to effectively attend to the clean conditioning features to guide the denoising of the noisy latent $z_t$.

This design philosophy is strongly supported by recent, concurrent research that analyzes the internal representations of diffusion models:

- The work on CleanDIFT (Stracke et al., 2025) directly argues that adding noise to images before feature extraction is a performance bottleneck that harms feature quality. Their entire method is built on the same premise as our $F_E$: that extracting features from clean images leads to superior performance without needing task-specific timestep tuning.

- Furthermore, ConceptAttention (Helbling et al., 2025) demonstrates that the internal representations of DiTs are highly interpretable and correspond to semantic concepts, particularly at early timesteps. This validates our choice to use $t = 0$ features, as they represent the purest and most semantically meaningful form of this information.

To validate our design choice, we conducted an ablation study comparing our method with the variant where the feature extractor $F_E$ and the denoising $F_D$ use the same synchronous timestep $t$. The results on Dress Code are presented in Table 6. As shown in the table, our proposed method with asynchronous timesteps significantly outperforms the synchronous variant across the majority of the metrics. This result provides strong empirical evidence for the value of clean conditioning and confirms the effectiveness of our proposed Multimodal Hybrid Attention.

Table 6: Additional ablation study results on the Dress Code dataset.

| | All | | | | | | Upper-body | | Lower-body | | Dresses | |
|---|---|---|---|---|---|---|---|---|---|---|---|---|
| | SSIM ↑ | PSNR ↑ | LPIPS ↓ | DISTS ↓ | FID ↓ | KID ↓ | DISTS ↓ | FID ↓ | DISTS ↓ | FID ↓ | DISTS ↓ | FID ↓ |
| *Effect of Varying λ Parameter* | | | | | | | | | | | | |
| $\lambda = 0.0$ (w/o g. aligner) | **76.01** | 12.85 | **30.84** | 20.63 | 5.91 | 0.78 | 21.77 | 11.26 | 22.26 | 14.22 | 17.86 | 11.86 |
| $\lambda = 0.25$ | 74.29 | 12.48 | 33.68 | 19.41 | 6.42 | 0.89 | **19.65** | **10.77** | 21.86 | 16.53 | 16.73 | 11.38 |
| $\lambda = 0.5$ *(Ours)* | 75.95 | **12.90** | 31.46 | 18.66 | 5.74 | 0.65 | 19.75 | 10.94 | **19.57** | **13.83** | **16.67** | 11.29 |
| $\lambda = 0.75$ | 71.93 | 11.71 | 37.03 | 20.35 | 7.81 | 1.41 | 20.11 | 11.09 | 23.85 | 21.12 | 17.10 | 11.69 |
| $\lambda = 1.0$ | 71.76 | 11.72 | 37.21 | 20.45 | 7.78 | 1.39 | 20.07 | 11.33 | 24.11 | 20.59 | 17.17 | 11.79 |
| *Effect of Varying the DiT Block i Used for $\mathcal{L}_{align}$* | | | | | | | | | | | | |
| $i = 6$ | 72.11 | 11.44 | 38.00 | 20.62 | 8.66 | 1.76 | 21.21 | 12.36 | 23.25 | 22.15 | 17.42 | 12.49 |
| $i = 8$ *(Ours)* | 75.95 | **12.90** | **31.46** | **18.66** | **5.74** | **0.65** | **19.75** | **10.94** | **19.57** | **13.83** | **16.67** | **11.29** |
| $i = 12$ | 75.30 | 12.59 | 32.71 | 19.13 | 6.48 | 0.87 | 20.41 | 11.86 | 20.01 | 15.40 | 16.98 | 11.72 |
| $i = 18$ | **76.16** | 12.57 | 31.66 | 19.17 | 6.87 | 1.16 | 20.27 | 12.37 | 19.86 | 15.52 | 17.38 | 12.28 |
| *Effect of Asynchronous Timestep Conditioning* | | | | | | | | | | | | |
| w/ same $t$ in $F_E$ and $F_D$ | **77.70** | 12.66 | 32.69 | 22.41 | 9.78 | 2.30 | 23.98 | 17.85 | 21.29 | 17.83 | 21.95 | 17.52 |
| w/ $t = 0$ in $F_E$ *(Ours)* | 75.95 | **12.90** | **31.46** | **18.66** | **5.74** | **0.65** | **19.75** | **10.94** | **19.57** | **13.83** | **16.67** | **11.29** |

Table 7: Comparison of CLIP vs. SigLIP 2 as vision encoder for person encoding. ↑ indicates higher is better, ↓ lower is better.

| Method | SSIM ↑ | PSNR ↑ | LPIPS ↓ | DISTS ↓ | FID ↓ | KID ↓ |
|---|---|---|---|---|---|---|
| TEMU-VTOFF w/ CLIP | 75.95 | 12.90 | 31.46 | **18.66** | 5.74 | 0.65 |
| TEMU-VTOFF w/ SigLIP 2 | **76.62** | **14.33** | **28.10** | 18.77 | **5.08** | **0.53** |

**Generalization with Stronger Vision Encoders.** We replace our CLIP vision encoder with a more powerful SigLIP 2 (Tschannen et al., 2025). We adopt the ViT-g 16 model and retrained our architecture with an additional MLP $f_{\mathrm{MLP}} : \mathbb{R}^{1536} \to \mathbb{R}^{2048}$ to project SigLIP 2 output dimension into the SD3 input space. The results are presented in Table 7. As noted in TryOffDiff (Velioglu et al., 2024), employing a stronger vision encoder improves the final performance. Our experiments further validate this finding. This improvement is due to the better capacity of SigLIP 2 at extracting more fine-grained features. As reported in (Tschannen et al., 2025), this model is trained with a binary contrastive loss that processes each text-image pair separately, thus preventing information corruption from different image-text pairs. Moreover, fine-grained details are enhanced with a self-distillation loss and masked prediction. Finally, this ablation further demonstrates that our core contribution lies in our dual-DiT architecture because this design scheme can be improved with plug-and-play modules, unlike the architectural alternatives that underperform in the same setting.

## C   ADDITIONAL QUALITATIVE RESULTS

We report an extended version of the qualitative results presented in our main paper. Specifically, additional visual comparisons between TEMU-VTOFF and competitors are shown in Fig. 9 and Fig. 10, on sample images from Dress Code (Morelli et al., 2022) and VITON-HD (Choi et al., 2021), respectively. Moreover, Fig. 11 presents additional ablation results to analyze the impact of textual and mask conditioning. Finally, we include in Fig. 12 the full set of inputs used for generating the target garment, including the model input, the segmentation mask, and the textual caption.

## D   USER STUDY

To complement our quantitative analysis and address the limitations of automated metrics in capturing fine-grained texture details, we conduct a human perceptual study.

**Experimental Setup.** We recruited 42 distinct participants to evaluate the visual quality of the generated garments. The study followed a pairwise comparison protocol. For each trial, participants were presented with the input person image and two generated garment results: one from TEMU-VTOFF and one from a competitor (randomly selected from MGT (Velioglu et al., 2025) or Any2AnyTryon (Guo et al., 2025)). The position of the images (*i.e.*, left/right) was randomized to prevent bias. Participants were asked to select the image that best represented a high-fidelity, in-shop

Table 8: Pairwise comparison showing the percentage of times participants preferred TEMU-VTOFF over the competing method. "Not Sure" indicates cases where participants found the quality of generated images indistinguishable.

| Comparison | Ours Wins (%) | Not Sure (%) | Competitor Wins (%) |
|---|---|---|---|
| TEMU-VTOFF vs. MGT | **75.77** | 7.85 | 16.38 |
| TEMU-VTOFF vs. Any2AnyTryon | **77.74** | 5.64 | 16.62 |

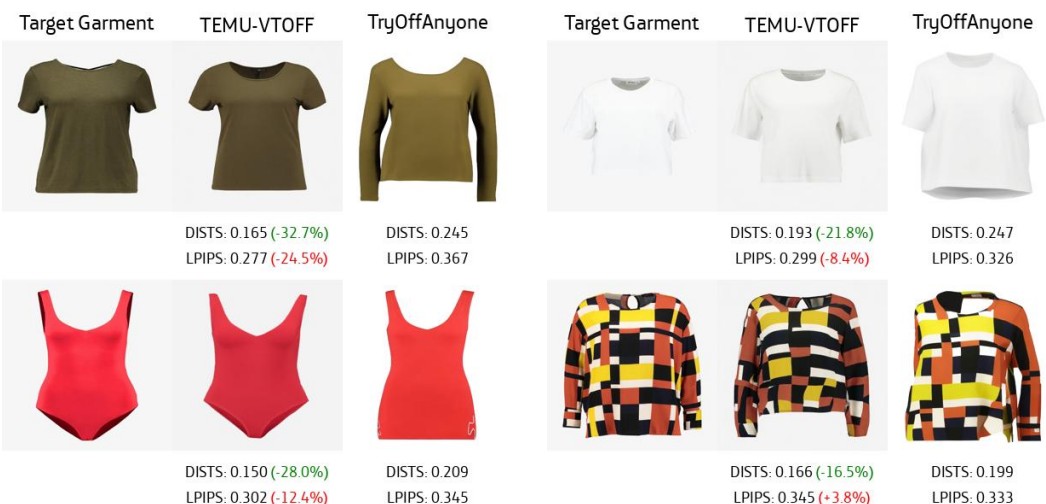

Figure 8: Additional qualitative results of TEMU-VTOFF and competitors on VITON-HD (Choi et al., 2021), with per-sample metrics. DISTS emphasizes structural differences between images better than LPIPS, confirming its higher correlation with human judgments.

version of the garment worn by the model, considering texture preservation, structural integrity, and overall realism.

**Results.** We collected a total of 1,920 pairwise judgments. The results, summarized in Table 8, demonstrate a strong preference for our method. TEMU-VTOFF outperforms MGT with a win rate of 75.77% and Any2AnyTryon with a win rate of 77.74%. These results strongly corroborate our quantitative analyses (particularly the DISTS and FID scores), confirming that TEMU-VTOFF produces results that are perceptually superior to state-of-the-art methods.

## E    DISCUSSION AND LIMITATIONS

Our method demonstrates strong performance and generalization, yet it inherits some inner problems of foundational models such as Stable Diffusion 3 (Esser et al., 2024). Although we improve the rendering of large logos and text, the model still struggles with fine-grained details, including complex texture patterns, small printed text, and the correct reproduction of small objects such as buttons. Moreover, as mentioned in the main paper, reconstruction is less reliable for lower-body garments than for upper-body items or dresses, likely due to class imbalance in the Dress Code dataset. For completeness, we show a set of failure cases in Fig. 13 and Fig. 14, on sample images from Dress Code (Cui et al., 2021) and VITON-HD (Lee et al., 2022), respectively.

We further analyze how the adopted perceptual metrics correlate with the presence of visual artifacts in the generated images (Fig. 8). While quantitative comparisons are reported as averages over the full test set, inspecting per-sample metric values is particularly informative in VTOFF, as it always lives in a paired setting. Edge cases, such as missing garment components or incorrect structural details, are often critical, and VTOFF naturally provides paired person-garment samples. In this context, LPIPS and DISTS play an important role, as both measure image-to-image distances. It is therefore essential to verify that these metrics respond reliably to detail discrepancies and appropriately penalize weaker baselines.

For each example in Fig. 8, we display three images: the ground-truth target, the output of TEMU-VTOFF, and the output of TryOffAnyone (Xarchakos & Koukopoulos, 2024). We present four representative comparisons. The first two (columns 1-3) show cases where TEMU-VTOFF preserves fine garment details that are lost by the competitor. The remaining two (columns 4-6) contrast accurate samples from our method with misaligned or rotated outputs produced by TryOffAnyone. For each pair, we report the corresponding per-sample DISTS and LPIPS values, along with the percentage improvement of our results over those of TryOffAnyone. In situations where a clear qualitative gap exists, DISTS consistently reflects the expected difference, whereas LPIPS often fails to penalize severe distortions and occasionally even assigns worse scores to the better-performing method (*e.g.*, row 2, column 5). These observations provide empirical evidence supporting our choice to include DISTS as part of the overall evaluation protocol. This observation is consistent with findings from DreamSim (Fu et al., 2023), as discussed in Sec. A.2.

## F   LLM USAGE

In this work, we employ LLMs (specifically Qwen2.5-VL) to extract garment-related textual descriptions, which serve as conditioning signals for generation. Beyond this, LLMs were employed solely for minor language refinement. They did not contribute to the design of experiments, the analysis of results, or the generation of scientific content.

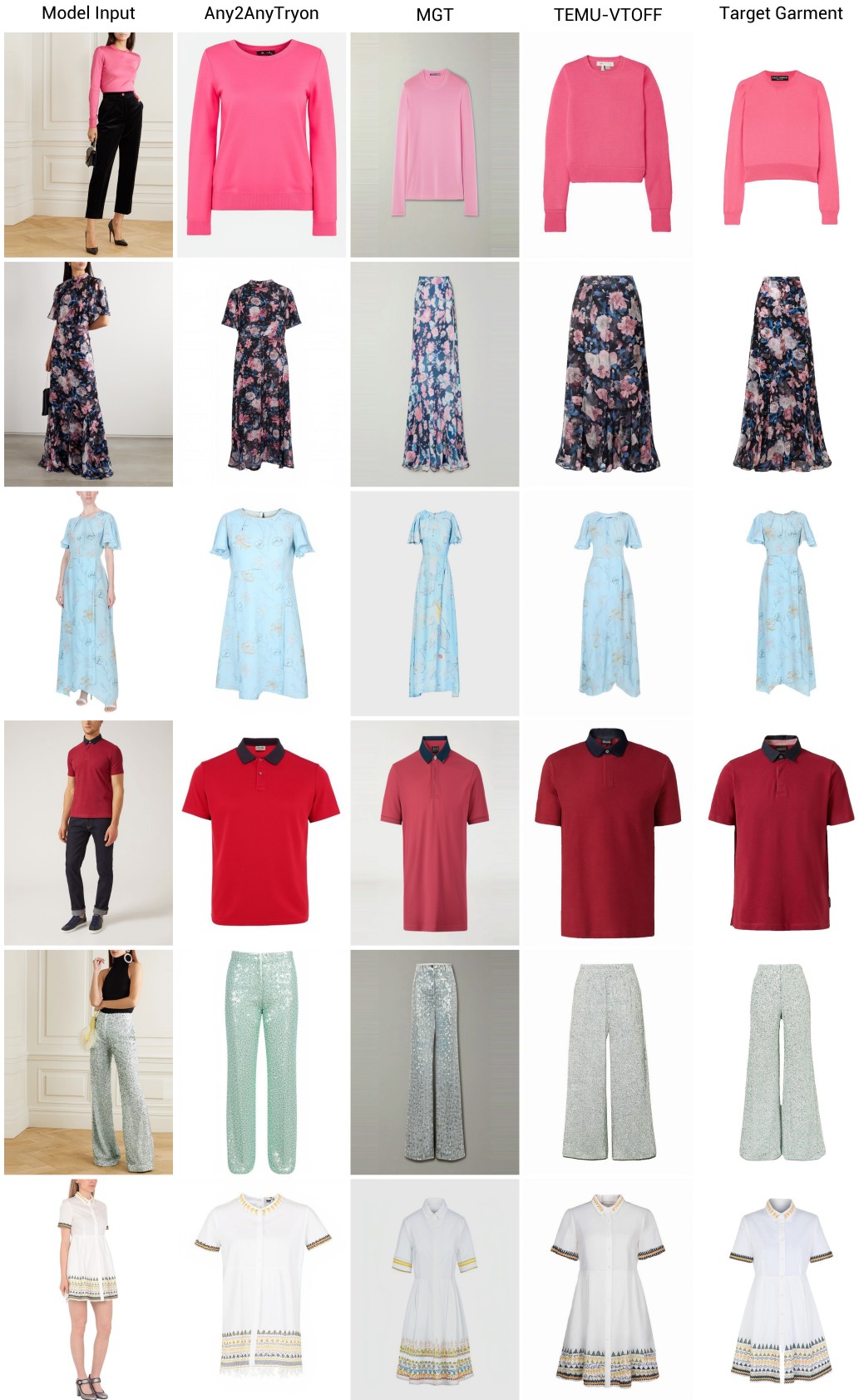

Figure 9: Additional qualitative results of TEMU-VTOFF and competitors on Dress Code (Morelli et al., 2022).

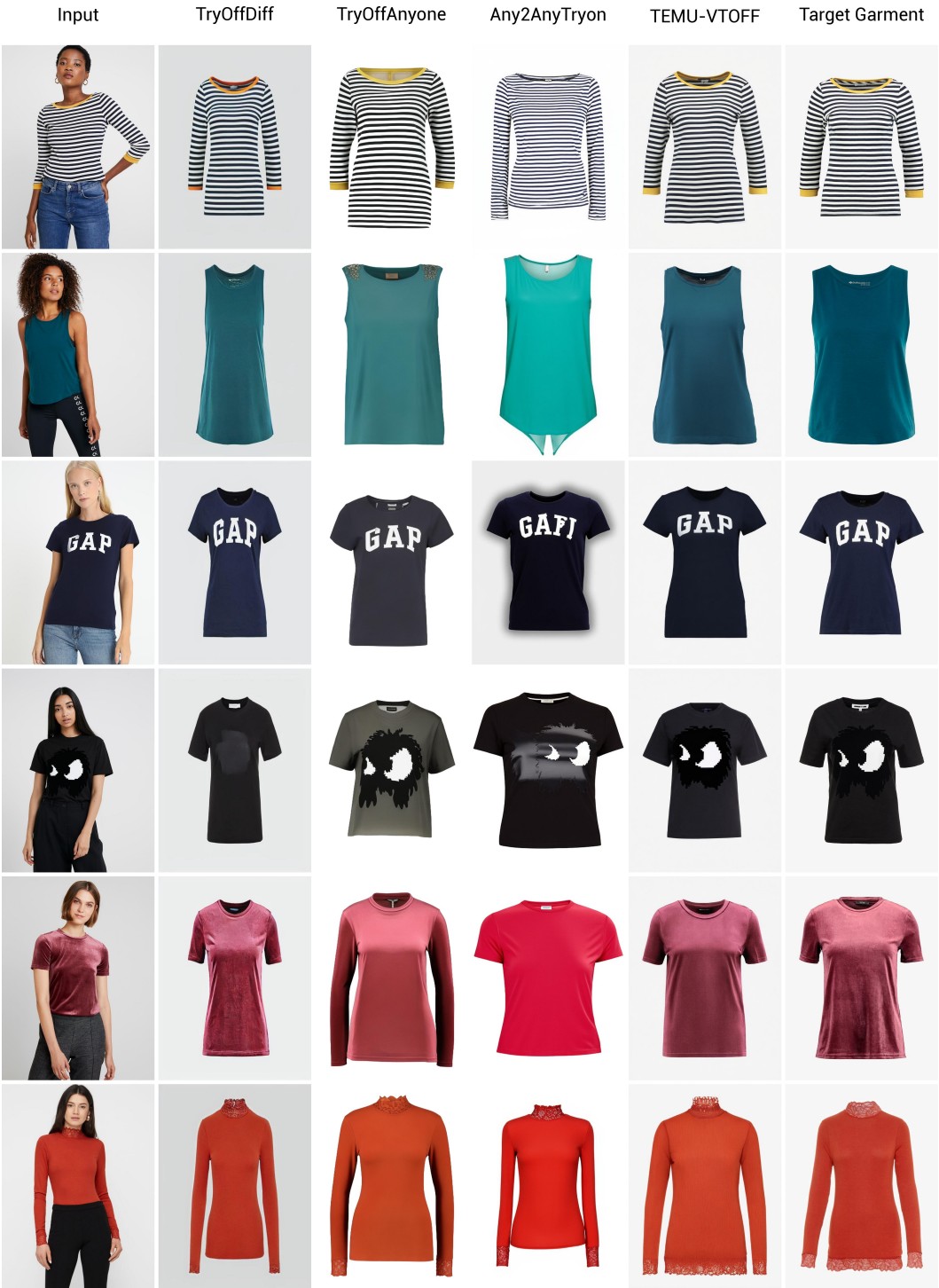

Figure 10: Additional qualitative results of TEMU-VTOFF and competitors on VITON-HD (Choi et al., 2021).

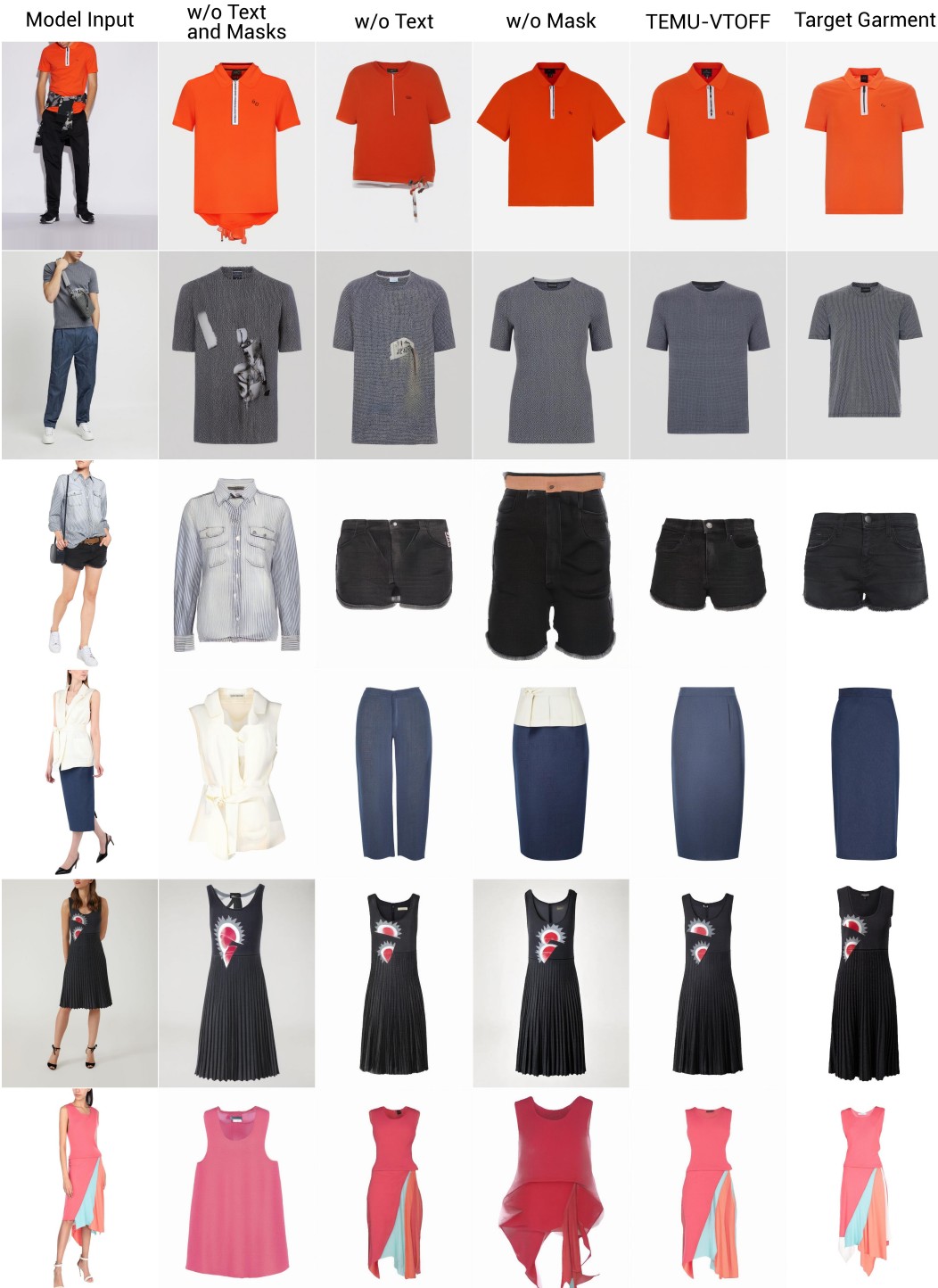

Figure 11: Additional qualitative results showing the contribution of each component in TEMU-VTOFF on Dress Code (Morelli et al., 2022) images.

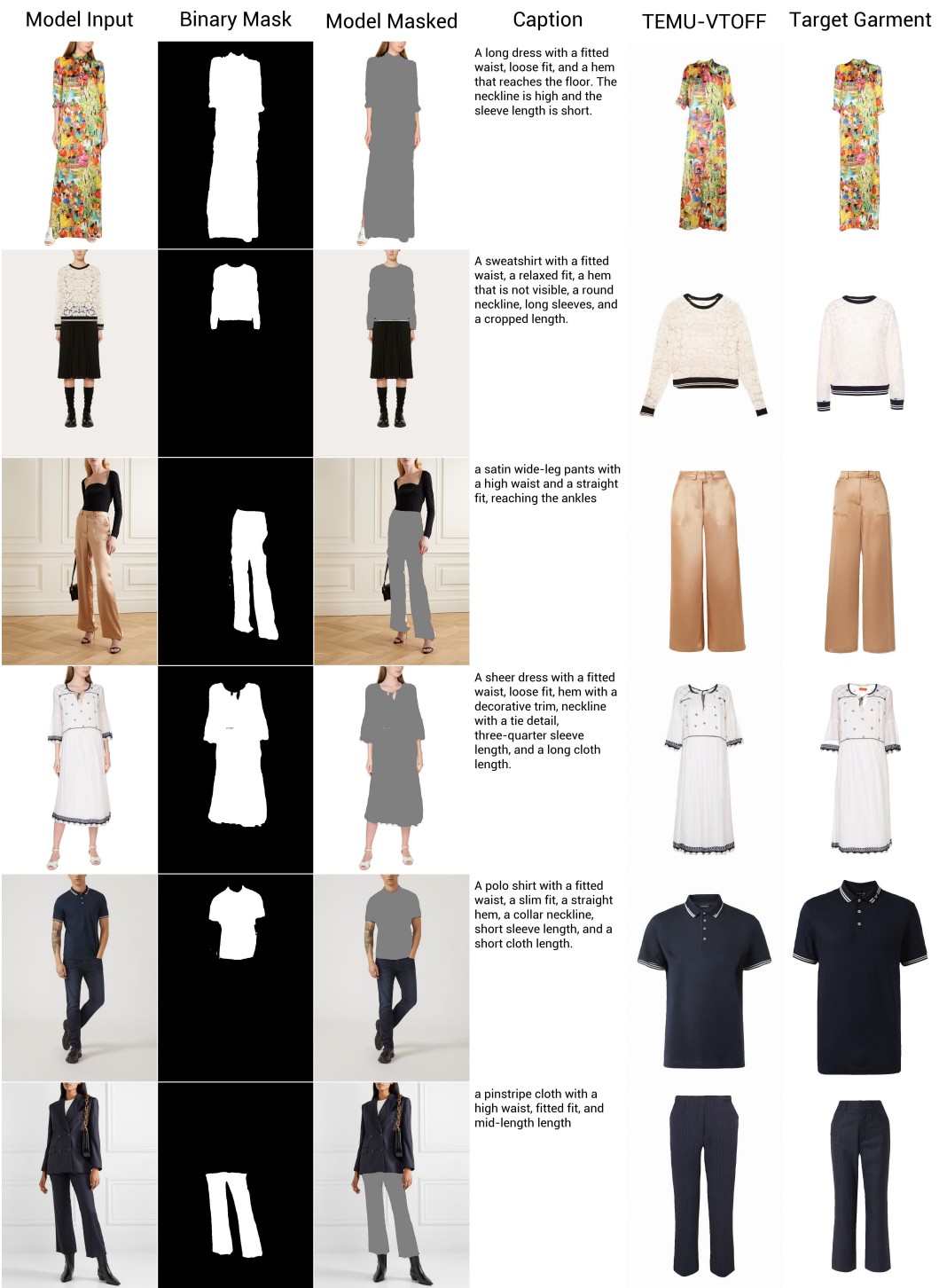

Figure 12: Inputs used to generate the target garment with TEMU-VTOFF, using sample images from Dress Code (Cui et al., 2021)

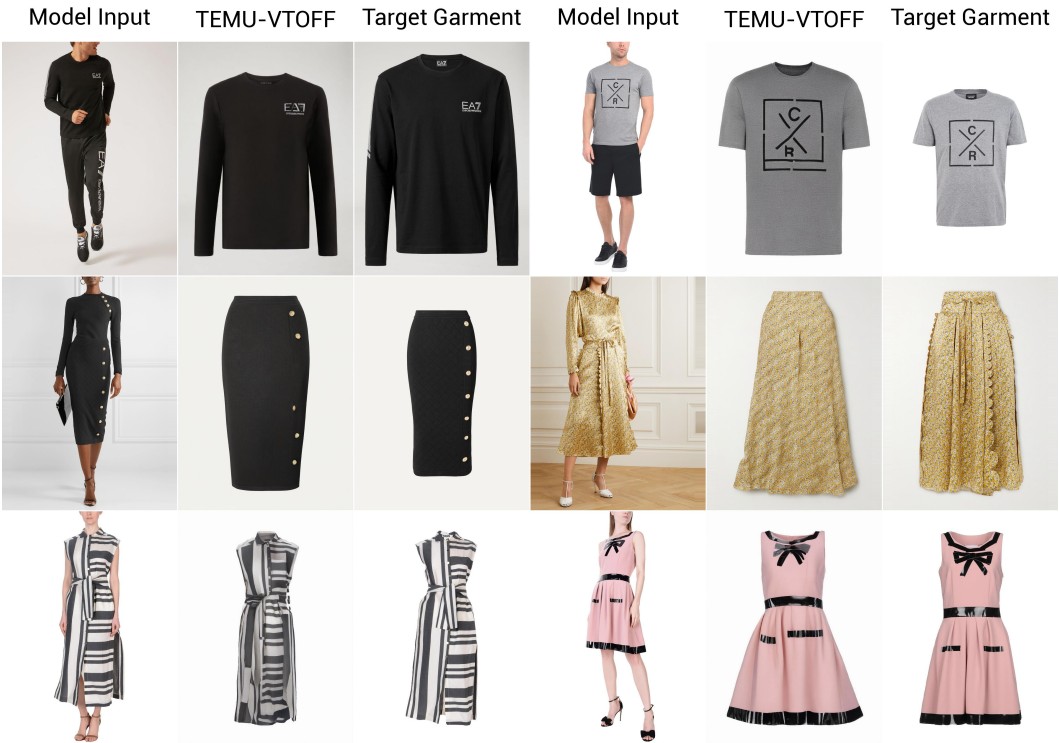

Figure 13: An overview of failure cases on the Dress Code (Morelli et al., 2022) dataset.

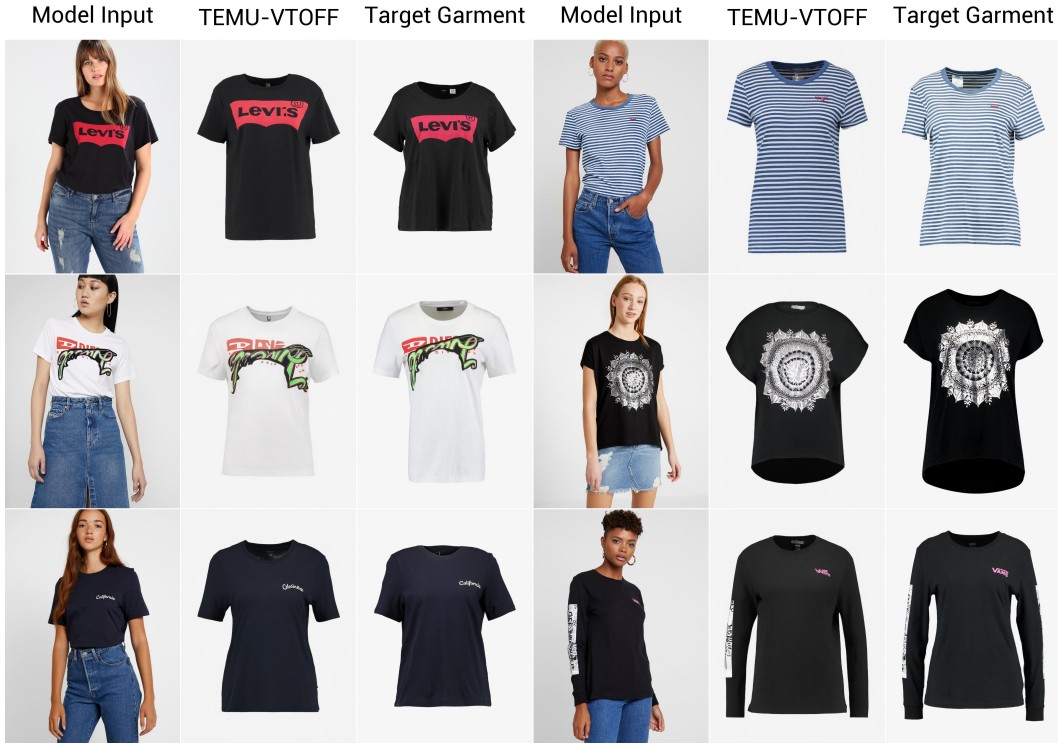

Figure 14: An overview of failure cases on the VITON-HD (Choi et al., 2021) dataset.

