# OpenReview forum: "Inverse Virtual Try-On: Generating Multi-Category Product-Style Images from Clothed Individuals"
_ICLR.cc/2026/Conference — ICLR 2026 Poster_

### Official Review · Reviewer_nzHz · 2025-10-20

**Soundness:** 2
**Presentation:** 3
**Contribution:** 2
**Rating:** 4
**Confidence:** 4

**Summary:**

This paper focuses on the virtual try-off task, which generates product images from clothed human images. The authors propose to use both visual and text cues for a richer feature representation from the clothing. In addition, an alignment loss is introduced to refine garment textures by matching DiT features with DINOv2 features. Results on two benmarks show the proposed method ourperforms existing approches.

**Strengths:**

1) The motivatio of introducing the alignment loss is clear.

2) The paper is structured well and easy to follow.

3) The authors provide many visualizations of the results.

**Weaknesses:**

1) The first contribution states that the proposed method does not require category-specific pipelines. However, many similar work including MGT and Any2AnyTryon have achieved this, which makes it less of a valid contribution.

2) In the model design,it is unclear why choosing the eighth Transformer block to match the DINO features. What are the justifications for this design choice? If it's empirical,  can the authors provide ablations since this is a major contribution of the paper? For example, will the generated details benefit from choosing more blocks to match DINO features? Also, since the network predicts noise rather than clean image, will the alignment loss degrades the image quality if the last few transformer blocks are used in the loss function?

3) Evaluation is not sufficient. Common objective metrics like the ones reported in the tables are not very consistent with human perception, especially on evaluating texture details. The authors should provide human study results on the benchmark for a more comprehensive evaluation.

**Questions:**

1) How important is the first DiT model that is trained on reconstructing human images? The authors mentioned that the features are better aligned because the two DiTs have similar architecture. However, adding noise to the input human image at early stages in the VTOFF inference seems unnecessary and could be hurtful. Will using coarse-to-fine clean feature layers in DINO with simple adaptors achieve similar effects?

2) In Table 1, lower-body clothing shows significantly worse scores than the other two categories. I would like to see more analysis on the reasons behind it.

---

> ### Author Response · Authors · 2025-11-24
> **Response to Reviewer nzHz (1/2)**
>
> ### [W1] Novelty of Unified Architecture vs. Existing Multi-Category Methods
>
> We respectfully clarify that while other methods technically support multiple categories, our contribution lies in the unified architecture that handles them natively without explicit branching or adaptation.
>
> - MGT: Relies on explicit, hard-coded class embeddings to switch behaviors.
> - Any2AnyTryon: Is not a native VTOFF model; it is an in-painting model adapted via LoRA.
> - TEMU-VTOFF: We propose a single, unified model where the Multimodal Hybrid Attention seamlessly resolves category ambiguity using the text caption and mask as natural soft-guidance. This architectural simplicity allows us to outperform these "multi-category" baselines, proving that our unified approach yields superior generation quality.
>
> ---
>
> ### [W2] Justification for Feature Alignment Depth (8th Block)
>
> We appreciate the reviewer's insightful question regarding the alignment depth. Our choice of the 8th block is driven by both theoretical findings in recent literature and our own empirical ablations.
>
> As noted in REPA [1], aligning early-to-mid layers (specifically around layer 8) is sufficient and often superior to aligning later layers. This is because early layers capture the structural and semantic essence of the image (which DINOv2 excels at), whereas later layers focus on predicting the high-frequency noise required for the diffusion objective. For this reason, aligning the last blocks would indeed be counter-productive. The output of the network at late stages must predict noise ($\epsilon$) or velocity ($v$), which is semantically distant from the clean, noise-invariant features of DINOv2. Forcing alignment at the end clashes with the diffusion objective, degrading fine detail generation.
>
> To validate this for our VTOFF task, we conduct an ablation study aligning the 6th, 8th, 12th, and 18th blocks of the DiT. The results on the Dress Code dataset are reported below:
>
> |     Block     |   PSNR ↑  |   SSIM ↑  |  LPIPS ↓  |  DISTS ↓  |   FID ↓  |   KID ↓  |
> |:-------------:|:---------:|:---------:|:---------:|:---------:|:--------:|:--------:
> |  6            |   11.44   |   72.11   |    38.00   |   20.62   |   8.66   |   1.76   |
> |  8 (Ours)     | **12.90** |   75.95   | **31.46** | **18.66** | **5.74** | **0.65** |
> |  12           |   12.59   |    75.30   |   32.71   |   19.13   |   6.48   |   0.87   |
> |  18           |   12.57   | **76.16** |   31.66   |   19.17   |   6.87   |   1.16   |
>
> Regularization applied to block 8 achieves the best perceptual quality (lowest LPIPS, DISTS, FID). It strikes the perfect balance, enforcing structural consistency (from DINOv2) while leaving enough subsequent layers free to refine high-frequency textures. Instead, in later blocks (12 and 18), as hypothesized, the performance degrades (FID increases from 5.74 to 6.87). While SSIM remains high, the perceptual metrics worsen, confirming that constraining the later "noisy" layers with "clean" DINO features hampers the model ability to generate realistic fine details.
>
> We have added this ablation study to Table 6, reported in Sec. B.1 of the Appendix.
>
> *[1] Yu, et al. Representation Alignment for Generation: Training Diffusion Transformers Is Easier Than You Think. ICLR, 2025*
>
> ---
>
> ### [W3] User Study and Alignment with Human Perception
>
> We thank the Reviewer for raising this point. We have expanded our previous justification for the choice of our metrics and their correlation with human judgment, reported in Sec. A.2 of the Appendix. To further elaborate, according to DreamSim [1], we can deduce that DISTS has the highest correlation with human judgment compared with LPIPS, SSIM, and PSNR. Moreover, DISTS is designed to disentangle structural and textual features of the encoded image by computing the global mean of each image and cross-image correlations. This ensures to have a reliable metric to detect texture changes across image pairs. As noted by the Reviewer, we acknowledge that some challenging, fine-grained details may still lead to wrong scores, but without compromising the overall trend of the dataset.
>
> To validate our quantitative metrics, we have also conducted a user-study on the Dress Code dataset between our method, MGT and Any2AnyTryon. In particular, we collected a total of 1,920 results from 42 different people. In the table below, we report the win rate of our method against the other baselines. The results confirm that TEMU-VTOFF is consistently ranked top against both, thus validating our quantitative analysis.
>
> | Model     | Win Rate (%) | Not Sure (%) | Lose Rate (%) |
> |:-:|:-:|:-:|:-:|
> | TEMU-VTOFF vs. MGT       | 75.77        | 7.85         | 16.38         |
> | TEMU-VTOFF vs. Any2AnyTryon   | 77.74        | 5.64         | 16.62         |
>
>
> We have included the user study results in Sec. D of the Appendix.
>
> *[1] Fu et al., DreamSim: Learning New Dimensions of Human Visual Similarity using Synthetic Data, NeurIPS 2025*

---

> ### Author Response · Authors · 2025-11-24
> **Response to Reviewer nzHz (2/2)**
>
> ### [Q1] Clarification on Inference Conditioning and Noise
>
> There is a slight misunderstanding of our inference pipeline, which we are happy to clarify. We explicitly **do not** add noise to the conditioning input of $F_E$ during inference. As detailed in Appendix B and the methodology, we extract features from $F_E$ using a clean image at timestep $t=0$. We specifically validated this choice in Table 4 (Appendix B). Using $t=0$ (clean extraction) yields an FID of 5.74, whereas using noisy time-steps ($t>0$) degrades the results.
>
> ---
>
> ### [Q2] Analysis of Lower Body Performance Drop
>
> The performance drop in the lower-body category is primarily due to dataset imbalance, a common issue in fashion datasets. We conduct our multi-category experiments on Dress Code [1], which has a total of three categories, including ~29k full body items (e.g. dresses), ~15k upper body items (e.g. t-shirts,) and ~8k lower body items (e.g. pants). This results in worse model performances on the least represented class. We have clarified this in both the main paper (cf. Sec. 4.1) and in the Appendix (cf. Sec. E).
>
> *[1] Morelli et al., Dress Code: High-resolution multi-category virtual try-on. ECCV, 2022*

---

### Official Review · Reviewer_Dtrc · 2025-10-29

**Soundness:** 3
**Presentation:** 3
**Contribution:** 3
**Rating:** 4
**Confidence:** 4

**Summary:**

The paper studies the inverse of virtual try-on, i.e., try-off. Given a photo of a person wearing clothes, the goal is to generate a standardized product-style image of the garment. This paper proposes TEMU-VTOFF, a dual DiT architecture with a feature extractor for the dressed person and a generator that attends to image, text, and mask tokens through a multimodal hybrid attention. A garment alignment module encourages high-frequency detail fidelity by aligning internal tokens to a frozen vision backbone. Experiments on Dress Code and VITON-HD report state-of-the-art results with ablations on text, mask, extractor features, and the alignment module.

**Strengths:**

- The task definition is with practical value. Inverse try-on is useful for catalog data creation and dataset enhancement. Multi-category handling in a single pipeline is appealing.

- The dual DiT setup and multimodal hybrid attention integrate signals from image, text, and mask in a straightforward and scalable way.

- Solid results on Dress Code. The method shows consistent improvements on distributional and perceptual metrics, with ablations that isolate design choices.

**Weaknesses:**

(1) Mixed gains on VITON-HD. On VITON-HD the improvements are minor or mixed. For example, LPIPS is **22.50** for One Model for All vs **28.44** for TEMU-VTOFF (LPIPS lower is better, so this favors the baseline), while DISTS is **19.20** vs **18.04** (lower is better, so this favors TEMU-VTOFF). This suggests the gains are not uniform across metrics or categories. A deeper per-category analysis is needed.

(2) Metric suitability and tradeoffs. The ablations imply the garment alignment module can trade off visual quality and alignment. Since paired ground truth exists, full-reference metrics like SSIM and LPIPS are directly meaningful. FID measures distributional similarity and can be less diagnostic in a paired setting. Please justify metric choices, add SSIM and PSNR consistently, and explain which metrics correlate with human preference for this task.

(3) Generalization and robustness. No cross-dataset experiments are reported. A simple but informative test is to train on VITON-HD and test on Dress Code, and vice versa, to evaluate robustness and domain shift.

(4) Qualitative artifacts. Several provided examples contain noticeable errors, which should be acknowledged and analyzed:
- Row 1, example 1: sleeves unexpectedly longer; skirt shows an unnatural shadow at the waist.
- Row 1, example 4: garment material looks inconsistent with the source.
- Row 2, example 2: button count and style differ from the source.
- Row 2, example 3: top body length should exceed sleeve length, but the result shows a shorter top.
- Row 2, example 4: trouser color shifts; a bow appears at the waistband that may reflect dataset bias.

A failure-mode analysis with per-category statistics would help.

(5) Broader utility not demonstrated. The paper motivates VTOFF as a tool for data generation, but does not show that the generated product images improve downstream tasks. A small study showing that try-off generated pairs improve try-on training or retrieval would make the case stronger.

(6) Notation and clarity.
Line 189: 𝑓=8 is used but f is never defined. I guess z_t  should be denoted as  (H/f)×(W/f)×3 if the latent has spatial downsampling by
𝑓. Please fix the symbol table and all affected equations.

**Questions:**

- Cross-dataset generalization. Can you report train-on-VITON-HD test-on-Dress-Code and the reverse, for both global metrics and per-category breakdowns?

- Metric justification. Given paired ground truth, why prioritize FID or KID? Please add SSIM and PSNR consistently and analyze correlations with human preference. If possible, add a small human study.

- Tradeoff in garment alignment. Can you quantify the quality–alignment tradeoff across a sweep of alignment strengths and report the setting used in the main results?

- Downstream utility. Try using TEMU-VTOFF to synthesize product tiles from arbitrary web or in-the-wild images to create pseudo-pairs for try-on training. Does this improve a standard VTON model’s accuracy or user preference?

- Caption provenance. How are text captions produced and sanitized to avoid leaking color or pattern attributes beyond the intended structure-only template?

**Details Of Ethics Concerns:**

There is potential for IP concerns when reconstructing designer items. Please document dataset licenses, prompt and caption sources, SD3 license compliance, and any safeguards against misuse.

---

> ### Author Response · Authors · 2025-11-24
> **Response to Reviewer Dtrc (1/4)**
>
> ### [W1] Performance Analysis on VITON-HD and Metric Selection
>
> We acknowledge the mixed signals in VITON-HD metrics but we believe they highlight the specific strengths of our method rather than inconsistency. In particular, VITON-HD contains only upper-body garments. As shown in Table 1 (Dress Code), our method performs better in complex categories (*e.g.* full-body garments). VITON-HD is a simpler task, leading to saturated performance where gains are harder to quantify. Moreover, in our experiments, we prioritize DISTS (where we achieve SOTA: 18.04 vs. 19.20) over LPIPS. Recent studies on human visual perception, such as DreamSim [1], demonstrate that DISTS correlates significantly better with human judgments, whereas LPIPS can be over-sensitive to pixel-level misalignment that does not affect perceived realism.
>
> *[1] Fu, et al. DreamSim: Learning New Dimensions of Human Visual Similarity using Synthetic Data. NeurIPS, 2023.*
>
> ---
>
> ### [W2 and Q2] Justification of Metrics and Trade-off Analysis (SSIM/PSNR)
>
> We appreciate the suggestion. We have computed PSNR and SSIM for all categories on Dress Code, and added PSNR to all tables of the paper. As shown in the tables below, TEMU-VTOFF achieves the best balance, securing the highest PSNR and SSIM in almost all categories while maintaining superior perceptual quality (FID/KID).
>
> While paired ground-truth exists, we decide to focus on distributional metrics because VTOFF is inherently a generative task. PSNR and SSIM penalize slight spatial misalignments, even if the reconstruction is photorealistic. FID and KID are essential to measure whether the generated output looks like a valid product image distribution, which is crucial for the downstream utility of the data.
>
> We report the per-category ablation tables of TEMU-VTOFF on Dress Code:
>
> **Dress Code - ALL**
>
> |         Method         |   PSNR ↑  |   SSIM ↑  |  LPIPS ↓  |  DISTS ↓  |   FID ↓  |   KID ↓  |
> |:----------------------:|:---------:|:---------:|:---------:|:---------:|:--------:|:--------:|
> | w/o text and masks     |   10.92   |   71.04   |   39.68   |   25.20   |   9.63   |   3.17   |
> | w/o text modulation    |   12.28   |   73.88   |   34.63   |   22.54   |   7.75   |   1.52   |
> | w/o fine-grained masks |   12.30   |   74.56   |   32.33   |   20.87   |   6.58   |   1.03   |
> | w/o Garment Aligner    |   12.85   | **76.01** | **30.84** |   20.63   |   5.91   |   0.78   |
> | TEMU-VTOFF (OURS)      | **12.90** |   75.95   |   31.46   | **18.66** | **5.74** | **0.65** |
>
> ---
> **Dress Code - Upper Body**
>
> |         Method         |   PSNR ↑  |   SSIM ↑  |  LPIPS ↓  |  DISTS ↓  |   FID ↓   |   KID ↓   |
> |:----------------------:|:---------:|:---------:|:---------:|:---------:|:---------:|:---------:|
> | w/o text and masks     |   11.16   |   71.35   |   39.35   |   23.71   |   19.75   |   3.22    |
> | w/o text modulation    |   11.89   |   72.50   |   38.52   |   24.02   |   13.49   |   2.33    |
> | w/o fine-grained masks |   12.43   |   74.75   | **33.68** |   20.85   |   11.31   |   1.01    |
> | w/o Garment Aligner    |   12.49   | **74.92** |   33.81   |   21.77   |   11.26   |   0.87    |
> | TEMU-VTOFF (OURS)      | **12.51** |   74.54   |   34.48   | **19.75** | **10.94** | **0.76**  |
>
> ---
> **Dress Code - Lower Body**
>
> |         Method         |   PSNR ↑  |   SSIM ↑  |  LPIPS ↓  |  DISTS ↓  |   FID ↓   |   KID ↓   |
> |:----------------------:|:---------:|:---------:|:---------:|:---------:|:---------:|:---------:|
> | w/o text and masks     |    9.53   |   65.85   |   49.19   |   31.77   |   31.77   |   36.22   |
> | w/o text modulation    |   11.84   |   71.81   |   37.26   |   24.33   |   18.13   |   4.03    |
> | w/o fine-grained masks |   11.94   |   72.90   |   33.85   |   22.34   |   15.75   |   2.16    |
> | w/o Garment Aligner    | **12.18** | **74.28** | **32.93** |   22.26   |   14.22   |   2.09    |
> | TEMU-VTOFF (OURS)      |   12.14   |   73.94   |    34.60   | **19.57** | **13.83** | **2.04** |
>
> ---
> **Dress Code - Dresses**
>
> |         Method         |   PSNR ↑  |   SSIM ↑  |  LPIPS ↓  |  DISTS ↓  |   FID ↓   |   KID ↓  |
> |:----------------------:|:---------:|:---------:|:---------:|:---------:|:---------:|:--------:|
> | w/o text and masks     |   12.63   |   75.91   |   30.49   |   20.12   |   15.47   |   2.20   |
> | w/o text modulation    |   13.24   |   77.32   |   28.12   |   19.27   |   13.30   |   1.34   |
> | w/o fine-grained masks |   12.54   |   76.31   |   29.48   |   19.42   |   13.62   |   1.80   |
> | w/o Garment Aligner    |   13.95   |   78.82   |   25.78   |   17.86   |   11.86   |   0.92   |
> | TEMU-VTOFF (OURS)      | **14.36** | **79.39** | **24.32** | **16.67** | **11.29** | **0.59** |

---

> > ### Author Response · Authors · 2025-11-24
> > **Response to Reviewer Dtrc (2/4)**
> >
> > ### [W2 and Q2] Justification of Metrics and Trade-off Analysis (SSIM/PSNR) - Continue
> >
> > Moreover, we report the quantitative results on the Dress Code dataset, including the PSNR:
> >
> > **Dress Code - ALL**
> >
> > |       Method      |  PSNR ↑  |   SSIM ↑  |  LPIPS ↓  |  DISTS ↓  |   FID ↓  |   KID ↓  |
> > |:-----------------:|:--------:|:---------:|:---------:|:---------:|:--------:|:--------:|
> > | Any2AnyTryon      |   12.67  |   77.56   |   35.17   |   25.17   |   12.32  |   3.65   |
> > | MGT               |   11.99  | **77.77** |   35.37   |   27.28   |   13.47  |   5.28   |
> > | TEMU-VTOFF (OURS) |**12.90** |   75.95   | **31.46** | **18.66** | **5.74** | **0.65** |
> >
> > ---
> > **Dress Code - Upper Body**
> >
> > |       Method      |   PSNR ↑  |   SSIM ↑  |  LPIPS ↓  |  DISTS ↓  |   FID ↓   |   KID ↓  |
> > |:-----------------:|:---------:|:---------:|:---------:|:---------:|:---------:|:--------:|
> > | TryOffDiff        |   11.54   |   76.59   |   40.62   |   29.04   |   37.97   |   17.3   |
> > | Any2AnyTryon      |   12.27   |   76.61   |   38.99   |   25.78   |   15.77   |   3.22   |
> > | MGT               |   11.44   | **76.77** |    39.70   |   28.13   |   19.49   |   6.87  |
> > | TEMU-VTOFF (OURS) | **12.51** |   74.54   | **35.48** | **19.75** | **10.94** | **0.76** |
> >
> > ---
> > **Dress Code - Lower Body**
> >
> > |       Method      |   PSNR ↑  |   SSIM ↑  |  LPIPS ↓ |  DISTS ↓  |   FID ↓   |   KID ↓  |
> > |:-----------------:|:---------:|:---------:|:--------:|:---------:|:---------:|:--------:|
> > | Any2AnyTryon      | **12.42** | **78.15** |   34.72  |   25.87   |   30.06   |   12.01  |
> > | MGT               |   11.64   |   77.29   |   36.31  |     28.00 |   25.98   |   9.64   |
> > | TEMU-VTOFF (OURS) |   12.14   |   73.94   | **34.60** | **19.57**| **13.83** | **2.04** |
> >
> > ---
> > **Dress Code - Dresses**
> >
> > |       Method      |   PSNR ↑  |   SSIM ↑  |  LPIPS ↓  |  DISTS ↓  |   FID ↓   |   KID ↓  |
> > |:-----------------:|:---------:|:---------:|:---------:|:---------:|:---------:|:--------:|
> > | Any2AnyTryon      |   13.32   |   77.93   |    31.80   |   23.86  |    19.20  |   6.27   |
> > | MGT               |   13.09   |   79.26   |   30.11   |    25.70  |   19.09   |   5.74   |
> > | TEMU-VTOFF (OURS) | **14.36** | **79.39** | **24.32** | **16.67** | **11.29** | **0.59** |
> >
> > ---
> >
> > ### [W3 and Q1] Cross-Dataset Generalization Capabilities
> >
> > To evaluate robustness and domain shift, we conducted the suggested cross-dataset experiments. As shown below, TEMU-VTOFF consistently outperforms competitors even when trained on a different domain, confirming its strong generalization capabilities.
> >
> > **Train on Dress Code $\rightarrow$ Test on VITON-HD**
> > |       Method      |   PSNR ↑  |   SSIM ↑  |  LPIPS ↓  |  DISTS ↓  |   FID ↓   |   KID ↓   |
> > |:-----------------:|:---------:|:---------:|:---------:|:---------:|:---------:|:---------:|
> > |               MGT |   10.24   | **74.26** |   42.57   |   28.73   |   23.11   |   10.81   |
> > | TEMU-VTOFF (Ours) | **10.85** |   72.80   | **40.19** | **24.20** | **20.39** | **7.00**  |
> >
> >
> > **Train on VITON-HD $\rightarrow$ Test on Dress Code**
> >
> > |       Method      |   PSNR ↑  |   SSIM ↑  |  LPIPS ↓  |  DISTS ↓  |   FID ↓   |   KID ↓  |
> > |:-----------------:|:---------:|:---------:|:---------:|:---------:|:---------:|:--------:|
> > |        TryOffDiff |   11.50   | **75.33** |   44.64   |   32.14   |   41.91   |   21.78  |
> > |      TryOffAnyone |   10.52   |   71.96   |   47.14   |   27.54   |   24.45   |   9.84   |
> > | TEMU-VTOFF (Ours) | **11.51** |   73.36   | **39.74** | **23.84** | **18.63** | **6.31** |
> >
> > In both settings, TEMU-VTOFF achieves significantly better perceptual (LPIPS, DISTS) and distributional (FID, KID) scores. Notably, when testing on Dress Code after training only on VITON-HD, our model achieves an FID of 18.63 compared to 24.45 for TryOffAnyone, demonstrating superior ability to generalize to unseen garment types and poses. Note that we exclude Any2AnyTryon from this specific analysis, as it is trained on a mixture of datasets, including both Dress Code and VITON-HD, making the cross-dataset evaluation unfair.

---

> ### Author Response · Authors · 2025-11-24
> **Response to Reviewer Dtrc (3/4)**
>
> ### [W4] Analysis of Qualitative Artifacts
>
> We thank the Reviewer for the meticulous analysis. Following these comments, we have updated our trailer image. Below, we explain every single raised issue.
> - Row 1, example 1: Some samples in the dataset present longer sleeves than body part. The shadow is consistent with a top-down lighting.
> - Row 1, example 4: We denote that both the fashion training datasets do not provide ground-truth information about the material of a garment. Therefore, our method is very strong at retaining structural and appearance details, while it must generalize to materials with incomplete information about them.
> - Row 2, example 2: We thank the reviewer for pointing this out. We have added this as a potential limitation of our method, reported in Sec. E of the Appendix.
> - Row 2, example 3: We respectfully disagree with this. The person wears the top body garment with rolled-up sleeves. As it may lead to confusion, we have changed it in the revised version.
> - Row 2, example 4: The garment color is correct as sometimes the illumination varies between training garment data and training person data. We do agree that the model inherits some small biases from the training data, in this case the sample is correct because the bow is hidden in the person image. Indeed, we have replaced this sample in the revised version for better clarity.
>
> ---
>
> ### [W5 and Q4] Demonstration of Downstream Utility for VTON Training
>
> To demonstrate utility, we conduct a downstream experiment: training a Virtual Try-On (VTON) model using our generated data. In particular, for each person image in the upper- and lower-body categories, we generate the missing in-shop garment: the lower-body item for upper-body images and the upper-body item for lower-body images. This procedure augments the dataset with additional person-garment pairs generated by TEMU-VTOFF. We then train the state-of-the-art CatVTON [1] model (using SD3 as backbone) on the original and the augmented datasets with two separate trainings. We test the trained models on the Dress Code test set.
>
> As shown below, augmenting with TEMU-VTOFF generated images consistently improves VTON performance, proving that our generated images are high-quality and structure-preserving enough to serve as training signal. We have added this important experiment in Sec. 4.4 of the main paper.
>
> ## CATEGORY – ALL
>
> | Method | PSNR ↑ | SSIM ↑ | LPIPS ↓ | DISTS ↓ | FID ↓ | KID ↓ |
> |:------------------------------:|:---------:|:------:|:--------:|:---------:|:-------:|:-------:|
> | CatVTON (Dress Code Original)  | 23.03 | **90.65** | 7.12 | 9.18 | 4.57 | 1.34 |
> | CatVTON (Dress Code Augmented) | **23.36** | **90.65** | **7.09** | **9.08** | **4.15** | **1.15** |
>
> ---
>
> ## CATEGORY – Upper Body
>
> | Method | PSNR ↑ | SSIM ↑ | LPIPS ↓ | DISTS ↓ | FID ↓ | KID ↓ |
> |:------------------------------:|:---------:|:----------:|:---------:|:----------:|:---------:|:---------:|
> | CatVTON (Dress Code Original)  | 24.32 | **92.93** | **5.33** | 7.66 | 9.58 | 2.05 |
> | CatVTON (Dress Code Augmented) | **24.39** | 90.94 | **5.33** | **7.54** | **9.27** | **1.75** |
>
> ---
>
> ## CATEGORY – Lower Body
>
> | Method | PSNR ↑ | SSIM ↑ | LPIPS ↓ | DISTS ↓ | FID ↓ | KID ↓ |
> |:------------------------------:|:---------:|:----------:|:----------:|:----------:|:---------:|:---------:|
> | CatVTON (Dress Code Original)  | 24.44 | 91.46 | 6.03 | 7.84 | 9.60 | 1.71 |
> | CatVTON (Dress Code Augmented) | **24.50** | **91.48** | **5.95** | **7.63** | **9.02** | **1.39** |
>
> ---
>
> ## CATEGORY – Dresses
>
> | Method | PSNR ↑ | SSIM ↑ | LPIPS ↓ | DISTS ↓ | FID ↓ | KID ↓ |
> |:------------------------------:|:---------:|:------:|:----------:|:----------:|:---------:|:---------:|
> | CatVTON (Dress Code Original)  | 21.18 | **87.50** | 10.01 | 12.04 | 9.58 | 1.26 |
> | CatVTON (Dress Code Augmented) | **21.21** | **87.50** | **10.00** | **12.02** | **9.46** | **1.13** |
>
> ---
>
> ### [W6] Clarification of Notation and Spatial Dimensions
>
> We apologize for the omission. $f=8$ represents the spatial downsampling factor of the SD3 VAE. The latent $z_t$ has dimensions $(H/f) \times (W/f) \times 16$. We have updated the symbol table and equations in the manuscript to explicitly define $f$ and the resulting spatial dimensions.
>
> *[1] Chong et al., Catvton: Concatenation is all you need for virtual try-on with diffusion models, ICLR 2025*

---

> > ### Author Response · Authors · 2025-11-24
> > **Response to Reviewer Dtrc (4/4)**
> >
> > ### [Q3] Quantification of Alignment Trade-offs
> >
> > In the main paper, we employ $\lambda=0.5$. We quantified this tradeoff in Appendix B (Table 4), where we perform a sweep of $\lambda$ values on the Dress Code dataset. The results, summarized below, demonstrate that $\lambda=0.5$ provides the optimal balance:
> >
> > |        Method       | PSNR ↑ | SSIM ↑ | LPIPS ↓ | DISTS ↓ | FID ↓ | KID ↓ |
> > |:-------------------:|:------:|:------:|:-------:|:-------:|:-----:|:-----:|
> > | w/o Garment Aligner | 12.85 |**76.01** |  **30.84**|20.63  |  5.91 |  0.78 |
> > |     $\lambda$ = 0,25    | 12.48 |74.29 |  33.68  |  19.41  |  6.42 |  0.89 |
> > |  $\lambda$ = 0,5 (Ours) | **12.90** |75.95 |  31.46|**18.66**|**5.74**|**0.65**|
> > |     $\lambda$ = 0,75    | 11.71 |71.93 |  37.03  |  20.35  |  7.81 |  1.41 |
> > |     $\lambda$ = 1,0     | 11.72 |71.76 |  37.21  |  20.45  |  7.78 |  1.39 |
> >
> > - Low Alignment ($\lambda=0$): While this setting maintains high structural similarity (SSIM 76.01), it lacks the fine-grained texture consistency required for high-fidelity reconstruction, resulting in worse perceptual scores (DISTS 20.63).
> > - Optimal Balance ($\lambda=0.5$): This setting achieves the best perceptual metrics (lowest DISTS, FID, and KID). Notably, it recovers fine details without degrading the overall structure, as SSIM (75.95) remains comparable to the baseline.
> > - Excessive Alignment ($\lambda \ge 0.75$): At higher weights, the alignment loss conflicts with the diffusion objective, leading to a sharp degradation in both structural and perceptual quality.
> >
> > ---
> >
> > ### [Q5] Prevention of Attribute Leakage in Text Captions
> >
> > To prevent leakage, we use a strict template-based generation with Qwen2.5-VL (see Appendix A.3). The prompt explicitly forbids generating color, texture, or pattern descriptions ("It's forbidden to generate... colors... textures/patterns"). Furthermore, we only extract structural tags (e.g., "V-neck", "Long sleeve", "Maxi length"). This forces the model to learn texture and color transfer solely from the visual features ($F_E$), ensuring the text acts only as a structural scaffold.
> >
> > ---
> >
> > ### [Ethics] Licensing, IP Concerns, and Usage Safeguards
> > We strictly adhere to ethical guidelines. We use Dress Code and VITON-HD, which are licensed for non-commercial academic research. SD3 is used under the Stability AI Community License. Furthermore, our code release will include a license restricting use to academic/non-commercial purposes to mitigate IP risks regarding designer items. We clarify that the tool is intended for dataset curation, not for reproducing copyrighted designs for commercial sale.

---

> > ### Comment · Reviewer_Dtrc · 2025-11-28
> >
> > I truly thank the authors for the extensive additional experiments, including the new metric tables, the cross-dataset evaluations, and the CatVTON downstream study. These clearly improve the paper. At the same time, several of my earlier concerns are only partly resolved, mostly at the level of interpretation and positioning rather than raw numbers.
> > - **On the “VITON-HD is saturated” explanation:** From your own tables, the relative behavior between Dress Code and VITON-HD is mixed and depends on both the method and the metric. For example, for baselines like Any2AnyTryon, many metrics favor Dress Code and less favor VITON-HD. This pattern suggests that VITON-HD may not be easier in some respects. In other words, it does not by itself justify the statement that the benchmark is saturated so that gains are hard to quantify. In particular, it does not fully explain the mixed relative improvements against One Model for All on VITON-HD, especially the large LPIPS gap. I would still welcome a more concrete analysis on VITON-HD that explains in which cases TEMU-VTOFF underperforms in LPIPS and why.
> > - **Metric choice in a paired setting:** Using DISTS in addition to LPIPS is reasonable, and the reference to DreamSim supports the idea that DISTS can correlate well with human perception. However, in VTOFF each input has a paired ground truth. This makes the problem much closer to reconstruction than to free form text to image generation. In such a paired setting, full reference metrics that compare directly to the ground truth, e.g., LPIPS (not sensitive to the pixel shift) should naturally play a central role, while distributional metrics such as FID and KID are complementary.
> > In other words, the current justification that VTOFF is inherently generative, therefore FID and KID should be prioritized because full reference metrics are too sensitive to misalignment, feels somewhat at odds with the task definition. Fine detail mismatches such as wrong bow location, button count, or neckline shape are exactly the kind of errors that a try-off system ought to be penalized for. I would encourage a clearer positioning where full reference metrics are treated as primary in this paired setup, and FID or KID are used to capture additional distributional aspects. If you believe DISTS is more reliable than LPIPS for this task, a short correlation analysis or a set of qualitative examples where DISTS better matches human preference than LPIPS would make this point more convincing. Otherwise, the choice of metric risks looking like it is mainly driven by where your method scores best.
> > - **Downstream CatVTON improvements:** The CatVTON experiment is a valuable addition and it is good to see that TEMU-VTOFF generated tiles can indeed be used as training data. At the same time, the absolute gains reported in Table 5 are quite small in most metrics. For example, on the “All” category, DISTS improves only slightly, and SSIM and PSNR change only marginally. Similar small deltas appear in several sub-categories.
> > - **Qualitative examples and fine attribute errors:** My intention in listing the visual issues in the teaser examples was not to demand perfection, but to highlight that some of the main demonstration figures contain visible attribute errors on details that are important for try-off. For example, in Figure 4, the bow on the ground truth tile is clearly shifted to one side, while TEMU-VTOFF produces a bow at the center, and TryOffAnyone appears closer to the ground truth in this respect. In another example, the bow or tie shape on TryOffAnyone matches the worn garment more closely, whereas TEMU-VTOFF largely loses this detail. These are not just cosmetic differences; they are exactly the kind of garment attributes that the system is supposed to recover. Since these are showcase examples, such inconsistencies slightly weaken the qualitative narrative that TEMU-VTOFF is always more faithful. It could also be helpful to report the local LPIPS and DISTS values for these specific crops, connecting the qualitative discussion of bows and buttons to the metric discussion in Section 4.
> >
> >
> >
> > Overall, I acknowledge that the authors have put efforts into the rebuttal and have added several useful experiments. My main remaining concerns are about how to interpret these results, how to position the choice of metrics in a paired setting, and how the limitations are reflected in both the text and the figures.

---

> > > ### Author Response · Authors · 2025-12-01
> > > **Response to Reviewer Dtrc**
> > >
> > > We sincerely thank the reviewer for the constructive feedback and for recognizing the value of the additional experiments. We appreciate the opportunity to clarify the interpretation of the metrics and the positioning of our results.
> > >
> > > 1. **VITON-HD and the LPIPS Gap (vs. One Model For All).** A central limitation in providing a deeper, case-based analysis of our LPIPS underperformance against One Model For All on the VITON-HD dataset is that the authors have not released source code or model weights. Our comparison therefore relies exclusively on the numbers reported in their paper, and we are unable to generate paired predictions the test set of the dataset to identify specific failure patterns. We have revised the text to clarify this limitation while pointing to an additional qualitative analysis in Appendix E to explain the LPIPS variability. Further details on LPIPS are provided in point 4 of this answer.
> > >
> > > 2. **Metric Choice in a Paired Setting.** We fully agree with the reviewer’s characterization: VTOFF is a paired reconstruction task, and full-reference metrics should indeed play a central role. In the revised manuscript, we explicitly reposition these metrics (SSIM, PSNR, LPIPS, and DISTS) as our primary measures of fidelity to the ground-truth (*i.e.*, target garment), while treating distributional metrics (FID, KID) as complementary indicators of realism. We also highlight that we have consistently reported all these metrics (SSIM, PSNR, LPIPS, DISTS) alongside FID/KID in all tables.
> > > In addition to quantitative metrics, we have also reported a user study that confirm the effectiveness of our approach. In particular, from the user study reported in Table 8, human raters preferred our method over competitors >75% of the time (cf. Sec. D of the Appendix), supporting the reliability of the trends captured by the full-reference metrics.
> > >
> > > 3. **Downstream CatVTON Improvements (Magnitude of Gains).** We acknowledge that the absolute gains are modest. However, we view the *presence* of measurable improvements as the central finding in this setting. CatVTON was retrained with roughly 27k additional synthetic pairs, and the fact that these generations lead to consistent (albeit small) positive shifts, rather than degrading performance, acts as an important test of the structural reliability of our outputs. We expect larger gains to emerge with increased scaling of the augmented dataset and with lightweight filtering strategies to remove low-quality generations.
> > > It is also important to clarify that this augmentation strategy is feasible only when the person image contains more than one garment type (*e.g.*, a ground-truth upper-body item that enables generating the missing lower-body item, or vice versa). For the “dresses” category, the full-body garment does not provide a complementary item that can be recovered from the same person image, preventing us from leveraging an additional ∼15k samples for CatVTON training. As no extra dress-category examples were generated, the performance in this category remains essentially unchanged compared to the CatVTON model trained on the original Dress Code dataset.
> > >
> > > 4. **Qualitative Artifacts and Fine Attribute Errors.** We agree that artifacts like the "shifted bow" or "wrong tie shape" can be identified as reconstruction failures. However, we highlight that these are not the only discriminants for the quality of the generated image. For instance, in the first sample of the second row of Fig. 4, TryOffAnyone produces a garment that is noticeably shorter than the ground-truth, whereas our method correctly preserves the garment length, an error that is structurally significant but not captured by examining small local attributes alone. We additionally note that, on VITON-HD, our method achieves consistently better results across all reported metrics compared to TryOffAnyone.
> > > To more directly address the reviewer’s concern, we have added a per-sample analysis in Fig. 8 and a detailed discussion in Sec. E of the Appendix. Specifically, the reported analysis examines how paired metrics correlate with visually evident differences between generated images. Our findings support the use of DISTS as a complementary full-reference metric: compared to LPIPS, DISTS is more sensitive to structural mismatches and therefore penalizes methods with larger geometric or shape deviations more consistently. This aligns well with the types of errors observed in the qualitative comparisons.

---

### Official Review · Reviewer_izfa · 2025-11-01

**Soundness:** 3
**Presentation:** 3
**Contribution:** 3
**Rating:** 6
**Confidence:** 4

**Summary:**

- This paper introduces TEMU-VTOFF, a diffusion-based virtual try-off model that reconstructs standardized product-style garment images from photos of clothed individuals, addressing a task that is largely unexplored compared to traditional virtual try-on.

- This paper uses a dual DiT architecture where one Transformer extracts garment features from the person image and the other generates the clean in-shop garment image, enhanced with multimodal attention using text and masks.

- This paper employs a garment alignment module and novel supervision loss to preserve structure and fine-grained textures, achieving state-of-the-art results on VITON-HD and Dress Code.

**Strengths:**

- Purpose-built architecture for try-off instead of reversing VTON pipelines, enabling clean reconstruction across multiple garment categories (upper / lower / full-body).

- Multimodal hybrid attention improves disambiguation and detail preservation by combining visual features with textual descriptions.

- High image fidelity and alignment thanks to the garment aligner module, resulting in superior quality and consistency compared to existing methods.

**Weaknesses:**

- Your attempt to explore a new direction within the VITON domain is impressive. However, while VITON-HD uses full-body datasets, this paper uses datasets without faces. Is this because including faces would cause errors?

- Would VTOFF also work on more limited imagery such as VITON-CROP [1]? Since this work deals with real-world scenarios, I recommend including [1] in the references.

- It would also be helpful if the ablation study section were organized in a more intuitive manner.

[1] Kang, Taewon, et al. "Data augmentation using random image cropping for high-resolution virtual try-on (VITON-CROP)." arXiv preprint arXiv:2111.08270 (2021).

**Questions:**

Mentioned in the weaknesses.

---

> ### Author Response · Authors · 2025-11-24
> **Response to Reviewer izfa**
>
> ### [W1] Clarification on Dataset Usage and Face Handling
>
> There appears to be a misunderstanding regarding the datasets used, which we are happy to clarify. We used VITON-HD, which contains faces and Dress Code, which contains cropped faces (see inputs in Figure 3 and Figure 4). Our model does not require removing faces. The inputs to our model are standard photos of people (often including faces), and the presence of a face does not cause errors.
> Furthermore, our architecture uses a mask-aware Feature Extractor and Multimodal Hybrid Attention. These components guide the model to focus specifically on the garment region while effectively ignoring non-garment features like the face or background during the reconstruction process.
>
> ---
>
> ### [W2] Robustness on Limited Imagery (VITON-CROP)
>
> Yes, we expect TEMU-VTOFF to handle limited or cropped imagery (like VITON-CROP) effectively. Unlike warping-based methods that require full visibility to map pixels, our method is based on a generative Diffusion Transformer (DiT). It has strong generative priors (learned from the underlying SD3 backbone), allowing it to generate and complete missing parts of a garment based on the visible regions and the textual description (*e.g.*, "long dress").
>
> We agree that [1] is highly relevant for real-world robustness discussions. We have added the missing citation to the related work section.
>
> *[1] Kang, Taewon, et al. "Data augmentation using random image cropping for high-resolution virtual try-on (VITON-CROP)." arXiv preprint arXiv:2111.08270 (2021).*
>
> ---
>
> ### [W3] Improved Organization of Ablation Studies
>
> We appreciate the feedback on the presentation. In Table 3, we currently group ablations by component type (Conditioning Inputs vs. Architecture vs. Regularization) to isolate the specific contribution of each design choice. To improve readability in the final version, we have reorganized the ablation study. This more clearly illustrates the incremental performance gain of adding each module to the pipeline.

---

> > ### Comment · Reviewer_izfa · 2025-11-26
> >
> > Thanks for your comments. I will maintain my score.

---

> > > ### Author Response · Authors · 2025-11-27
> > > **Response to Reviewer izfa**
> > >
> > > Thank you for your prompt response. We hope our response has adequately addressed your concerns.
> > > If any remaining questions or points would benefit from further clarification, please let us know.
> > >
> > > The Authors

---

### Official Review · Reviewer_mTyd · 2025-11-01

**Soundness:** 3
**Presentation:** 4
**Contribution:** 3
**Rating:** 6
**Confidence:** 4

**Summary:**

This paper considers multi-garment Virtual Try Off. The proposed approach leverages a dual DiT architecture based on Stable Diffusion 3, where the first network serves as a feature extractor, and the second diffuses the garment itself.

The feature extractor is trained to diffuse the latents of the model image, and takes as its input the latent $z^t$, the encoded latents of the masked model image and the binary mask of the garment. The diffusion is conditioned on the CLIP embeddings of the original model image scaled and shifted with AdaLN.

The diffusion network is conditioned on intermediate outputs of the feature extractor and CLIP and T2 textual embeddings of garment captions obtained by Qwen2.5-VL. These are combined in the proposed  Multimodal Hybrid Attention module.

To further promote detail preservation, the intermediate features of the 8th internal block of the diffusion network is aligned with DINOv2 features.

Training is done in two stages: First the feature extractor is trained. In the second stage, only the diffusion network is trained, with the values of the feature extractor for timestep 0 serving as the conditioning for all of the timesteps of the diffusion process.

**Strengths:**

[S1] Good quantitative and qualitative results

[S2] A good ablation study justifying most of the design choices.

[S3] Well written and easy to follow.

**Weaknesses:**

[W1] A3 section of the appendix suggests that the garment captions are based on textual descriptions of the e-commerce garment image. This seems like a fundamental flaw as the original garment caption is not going to exist for samples in the wild where the ground truth is not going to be known. This presents information about test directly seeping into the inference process.

[W2] Some unclear implementation details. See questions.

[W3] A couple of additional ablations would be useful. e.g. Velioglu et al. (BMVC, 2025) report better results with SigLip encoding of the conditioning image. Furthermore, why is it necessary to have two different textual embedders applied to the caption and concatenated?

**Questions:**

[Q1] Which encoder is used to encode the the masked person image in the feature extractor?

[Q2] What is $z_g$ in equation 7? Is the diffusion model in the latent space or the image space?

[Q3] Is garment aligner used during inference?

[Q4] How does this model perform on the samples not from the training dataset? How sensitive is it to the errors in the masking of the model picture?

---

> ### Author Response · Authors · 2025-11-24
> **Response to Reviewer mTyd (1/2)**
>
> ### [W1] Clarification on Inference Protocol and Data Leakage Concerns
>
> We apologize for the ambiguity in Sec. A.3 of the Appendix and appreciate the opportunity to clarify this critical point. **In our training-infence pipeline, there is no data leakage from the test set.**
> - **Training**: We generate captions from the ground-truth product images to ensure the model learns precise semantic correlations during optimization.
> - **Inference**: At test time, the ground-truth product image is not used. Instead, the caption is generated directly from the **input person image** using the same Qwen2.5-VL pipeline. This ensures that our inference process relies solely on the input person image, making the method fully applicable to "in-the-wild" scenarios where the product image is unavailable.
>
> We have revised Sec. A.3 to explicitly state the inference-time protocol.
>
> ---
>
> ### [W2] Clarification on Implementation Details
>
> We have addressed all implementation queries in the "Questions" section below. We have incorporated these clarifications into the manuscript to improve reproducibility.
>
> ---
>
> ### [W3] Impact of Stronger Vision Encoders (SigLIP 2) and Dual Text Embedders
> - **SigLIP 2 vs. CLIP**
> We thank the reviewer for suggesting incorporating a more powerful vision encoder than the original CLIP. We retrain our method using SigLIP 2 [1] as vision encoder for person features, with a trainable MLP to align the original SigLIP 2 dimension to the DiT expected input dimension.
>
> Below, we report an additional table that shows the improvement of our method employing SigLIP 2 compared to our original CLIP-based implementation:
>
> | Method                  | PSNR ↑  | SSIM ↑    | LPIPS ↓ | DISTS ↓ | FID ↓      | KID ↓       |
> |-------------------------|---------|-----------|---------|---------|--------    |-------      |
> | TEMU-VTOFF w/ CLIP      |	12.90    | 75.95    |   31.46 | **18.66** |	5.74     |	0.65      |
> | TEMU-VTOFF w/ SigLIP 2  |**14.33**| **76.62** | **28.10** |	18.77 |	**5.08** |**0.53**|
>
> As noted in TryOffDiff [2], employing a stronger vision encoder improves the final performance. Our experiments further validate this finding. This improvement is due to the better capacity of SigLIP 2 at extracting more fine-grained features. As reported in their paper, this model is trained with a binary contrastive loss that processes each text-image pairs separately, thus preventing information corruption from different image-text pairs. Moreover, fine-grained details are enhanced with a self-distillation loss and masked prediction. Finally, this ablation further demonstrates that our core contribution lies in our dual-DiT architecture because this design scheme can be improved with plug-and-play modules, unlike the architectural alternatives (such as single-DiT, see Table 3) that underperform in the same setting. We have added this additional ablation study in Sec. B of the Appendix.
>
> - **Dual Text Embedders (CLIP + T5)**: We follow the SD3 design philosophy [3], which demonstrates that these encoders are complementary:
>     - CLIP provides strong global visual-semantic alignment.
>     - T5 (LLM-based) captures complex structural details and fine-grained language understanding (*e.g.*, ensuring distinction between "short sleeve" and "three-quarter sleeve").
>
> Furthermore, our ablation in Table 3 ("w/o text modulation") already shows that removing text guidance degrades the performance. Both encoders are necessary to fully condition the complex multi-category generation.
>
> *[1] Tschannen, et al. SigLIP 2: Multilingual Vision-Language Encoders with Improved Semantic Understanding, Localization, and Dense Features. arXiv, 2025*
>
> *[2] Velioglu, et al. TryOffDiff: Virtual-Try-Off via High-Fidelity Garment Reconstruction using Diffusion Models. BMVC, 2025*
>
> *[3] Esser, et al. Scaling Rectified Flow Transformers for High-Resolution Image Synthesis. ICML, 2024*
>
> ---
>
> ### [Q1] Feature Extractor Encoding Details
>
> We clarify that the masked person image $x_m=\mathcal{E}(I_M)$ is encoded using the SD3 VAE encoder $\mathcal{E}$. Specifically, $x_M = \mathcal{E}(x_{model} \odot M)$, where $x_{model}$ is the person image and $M$ is the mask. This ensures that the masked person features reside in the same latent space (16 channels) as the noisy latent $z_t$, facilitating effective concatenation in the feature extractor input $z'_t$.
>
> ---
>
> ### [Q2] Clarification of Latent Space Notation
>
> In Eq.(7), $z_g$ denotes the **latent representation** of the target garment image $x_g$, obtained via the SD3 encoder $\mathcal{E}$: $z_g=\mathcal{E}(x_g)$. We have revised the notation to explicitly define $z_g=\mathcal{E}(x_g)$ and avoid confusion with pixel-space variables.

---

> > ### Author Response · Authors · 2025-11-24
> > **Response to Reviewer mTyd (2/2)**
> >
> > ### [Q3] Computational Cost of Garment Aligner at Inference
> >
> > No, the garment aligner is strictly a training-time component. It computes an auxiliary loss ($\mathcal{L}_{align}$) to enforce consistency between the DiT features and DINOv2 features. During inference, we only run the Dual-DiT backbone ($F_E$ and $F_D$). Consequently, the aligner adds zero computational cost at inference time. We have added this clarification in Sec. 3.3.
> >
> > ---
> >
> > ### [Q4] Cross-Dataset Generalization and Sensitivity to Masking Errors
> >
> > **Performance on Unseen Data:** We thank the reviewer for raising a well-known issue in the fashion domain. To demonstrate the effectiveness of our approach, and to evaluate robustness and domain shift, we conducted a cross-dataset experiments. As shown below, TEMU-VTOFF consistently outperforms competitors even when trained on a different domain, confirming its strong generalization capabilities on unseen data.
> >
> > **Train on Dress Code $\rightarrow$ Test on VITON-HD**
> > |       Method      |   PSNR ↑  |   SSIM ↑  |  LPIPS ↓  |  DISTS ↓  |   FID ↓   |   KID ↓   |
> > |:-----------------:|:---------:|:---------:|:---------:|:---------:|:---------:|:---------:|
> > |               MGT |   10.24   | **74.26** |   42.57   |   28.73   |   23.11   |   10.81   |
> > | TEMU-VTOFF (Ours) | **10.85** |   72.80   | **40.19** | **24.20** | **20.39** | **7.00**  |
> >
> >
> > **Train on VITON-HD $\rightarrow$ Test on Dress Code**
> >
> > |       Method      |   PSNR ↑  |   SSIM ↑  |  LPIPS ↓  |  DISTS ↓  |   FID ↓   |   KID ↓  |
> > |:-----------------:|:---------:|:---------:|:---------:|:---------:|:---------:|:--------:|
> > |        TryOffDiff |   11.50   | **75.33** |   44.64   |   32.14   |   41.91   |   21.78  |
> > |      TryOffAnyone |   10.52   |   71.96   |   47.14   |   27.54   |   24.45   |   9.84   |
> > | TEMU-VTOFF (Ours) | **11.51** |   73.36   | **39.74** | **23.84** | **18.63** | **6.31** |
> >
> > In both settings, TEMU-VTOFF achieves significantly better perceptual (LPIPS, DISTS) and distributional (FID, KID) scores. Notably, when testing on Dress Code after training only on VITON-HD, our model achieves an FID of 18.63 compared to 24.45 for TryOffAnyone, demonstrating superior ability to generalize to unseen garment types and poses. Note that we exclude Any2AnyTryon from this specific analysis, as it is trained on a mixture of datasets including both Dress Code and VITON-HD, making the cross-dataset evaluation unfair.
> >
> > We have added these experimental results in Sec. 4.3 of the main paper.
> >
> > **Sensitivity to Mask Errors:** The model is robust to moderate masking errors for two reasons:
> > 1. Training Dropout: During training, we randomly drop the mask input (probability $p=0.1$). This forces the model to learn to separate the garment from the person using semantic cues (from $F_E$ and Text) rather than relying solely on the mask boundaries.
> > 2. Real-World Noise: The training masks in Dress Code and VITON-HD are automatically generated and often contain imperfections (noise, missing regions). The model is trained on this noisy data, effectively learning to be robust to similar imperfections at inference time.

---

### Author Response · Authors · 2025-11-24
**Overall Response Summary**

We sincerely thank all reviewers and the AC for their time, constructive feedback, and thoughtful evaluation of our submission. We truly appreciate the care each reviewer put into both the strengths they identified and the concerns they raised.

We are encouraged that the reviewers broadly recognized the value and practicality of our work. In particular:
- **R1 (mTyd)** highlighted the strong quantitative and qualitative results, the comprehensive ablation study, and the clarity of presentation.
- **R2 (izfa)** appreciated that the architecture is purpose-built for virtual try-off, the effectiveness of the multimodal hybrid attention, and the high image fidelity enabled by the alignment module.
- **R3 (Dtrc)** acknowledged the practical relevance of the VTOFF task, the appeal of a single multi-category pipeline, and the solid Dress Code results supported by clean ablations.
- **R4 (nzHz)** noted the clarity of the motivation for the alignment loss, the overall organization of the paper, and the extensive visualizations.

We are grateful for these positive assessments, which confirm that the main contributions of **TEMU-VTOFF** are well-motivated and technically meaningful. Below, we respond point-by-point to each reviewer in their respective sections. In parallel, we have strengthened the manuscript through several key revisions, including:
1. **Clarifying the caption pipeline**, explicitly stating that inference relies solely on captions generated from the *input person image*, ensuring there is no test-time leakage.
2. **Adding new ablations and stronger backbones (e.g., SigLIP-2)** to validate the flexibility and complementarity of our design choices such as the dual text encoders and dual-DiT architecture.
3. **Expanding generalization evidence**, including cross-dataset train/test evaluations showing robustness under domain shift.
4. **Enhancing metric justification**, consistently reporting PSNR/SSIM, explaining why perceptual/distributional metrics are essential for this generative task, and adding a small human study to validate alignment with human preference.
5. **Improving clarity and reproducibility**, fixing notation issues (e.g., the VAE downsampling factor \(f\)) and reorganizing ablations for better readability.

These points summarize the main revisions; all additional clarifications and technical updates are addressed directly within the reviewer-specific responses below. For full transparency, we have highlighted every change in the revised manuscript in blue.

We sincerely appreciate the opportunity to further improve the submission, and any additional comments from the reviewers are welcome.

Best regards,

Submission 17076 authors

---

> ### Public Comment · ~Shuang_Liu10 · 2025-11-25
>
> Hello authors, thank you for the excellent work on TEMU-VTOFF.
>
> I have a concern similar to Reviewer mTyd regarding the source of captions during inference.
>
> In the paper, Sec. 3.2 emphasizes that textual structural guidance helps recover occluded garment regions (“This structural guidance enables the model to better capture the structure of the garment, leading to significantly more accurate outputs in occluded scenarios.”). This strongly suggests that the caption is generated from the model/person image, since only the person image contains such occlusions. However, the paper never explicitly states whether, during inference, the caption is generated from the person image. This ambiguity can affect reproducibility because caption content directly influences the hybrid attention module and consequently the garment reconstruction quality. Moreover, appendix B only describes how captions are generated from garment images during training. However, the paper does not provide an equivalent description for how captions are generated during inference.
>
> Thanks again for the strong contribution!

---

> > ### Author Response · Authors · 2025-11-26
> > **Response to Public Comment**
> >
> > Dear Commenter,
> >
> > Thank you for your positive feedback on TEMU-VTOFF and for highlighting this critical detail regarding reproducibility.
> >
> > You are correct in your observation. At inference time, the caption is generated directly from the input person image.
> >
> > As you noted, this is essential for two reasons:
> > - Practicality: In real-world "in-the-wild" scenarios, the ground-truth garment image is unavailable, so we must rely solely on the person image for structural guidance.
> > - Occlusion Handling: As mentioned in Sec. 3.2, generating the description from the person image allows the VLM (Qwen2.5-VL) to infer the likely structure of the garment based on visual cues, which then guides the reconstruction of occluded regions.
> >
> > We apologize for the ambiguity in the previous version. We have updated the manuscript (specifically Appendix A.3) to explicitly detail this inference protocol. The revised section now clearly distinguishes between the training phase (where captions are derived from ground-truth garment images to learn precise semantic correlations) and the inference phase (where captions are derived from person images to prevent data leakage). Furthermore, we have added the prompt used at inference time, as requested.
> >
> > Thank you again for helping us improve the clarity and reproducibility of our work.
> >
> > Best regards, The Authors

---

> > > ### Public Comment · ~Shuang_Liu10 · 2025-11-27
> > >
> > > Thank you for the clarifications. My concerns have been addressed.

---

### Author Response · Authors · 2025-12-01
**Summary of Reviewer Concerns, Responses and Manuscript Revisions**

We sincerely thank all reviewers and the AC for their time, constructive feedback, and thoughtful evaluation of our submission. As the discussion period concludes, we provide below a concise summary of the key concerns raised, our responses, and the corresponding revisions incorporated into the manuscript.

---

### Reviewer mTyd

*   **Potential data leakage in caption generation.** We clarified that inference relies strictly on captions generated from the input person image, ensuring no leakage at test time. **Sec. A.3 of the Appendix** has been updated to explicitly describe the captioning protocol and prompts used during inference.
*   **Need for stronger visual backbones.** We trained an additional variant of our model using SigLIP-2, demonstrating that the architecture generalizes well to more powerful visual encoders. The corresponding ablation, included in **Appendix B (Table 7)**, shows consistent performance improvements.
*   **Use of the garment aligner at inference time:** We revised **Sec. 3.3** to make clear that the garment aligner incurs no inference-time cost, as it is used solely during training to improve overall generation quality.

---

### Reviewer izfa

*   **Robustness on datasets with different image types.** We clarified that our masking mechanism effectively handles faces and cropped inputs (*e.g.*, VITON-CROP), and we have also incorporated this work into the related work discussion for completeness.
*   **Organization of ablation studies.** We reorganized **Table 3** to improve clarity and make the structure of the ablations easier to interpret.

---

### Reviewer Dtrc

*   **Metric suitability.** We clarified the central role of full-reference metrics (SSIM, PSNR, LPIPS, DISTS) relative to distributional metrics (FID, KID), and added the requested PSNR metric to **all tables** throughout the paper.
*   **Mixed LPIPS improvements:** We included an additional analysis in **Appendix E** discussing known limitations of LPIPS and justifying our use of DISTS as a complementary metric. For the mixed improvements observed in Table 2 (VITON-HD), we also clarified that the results for One Model For All are taken directly from the original paper, as neither code nor weights are publicly available, preventing a per-sample comparison of their method with ours.
*   **Cross-dataset generalization:** We added cross-dataset evaluations in **Sec. 4.3 (Table 4)**, training on VITON-HD and testing on Dress Code and vice versa. These results further confirm the generalization ability of our approach.
*   **Downstream utility:** We introduced a downstream experiment in **Sec. 4.4 (Table 5)**, showing that augmenting a virtual try-on training set with TEMU-VTOFF generations improves the performance of a state-of-the-art model (*i.e.*, CatVTON).
*   **Qualitative artifacts:** To complement our quantitative and qualitative analyses, we added a user study comprising 1,920 pairwise comparisons in **Appendix D (Table 8)**, where human raters preferred our method in more than 75% of cases.

---

### Reviewer nzHz

*   **Justification for the alignment depth (8th block):** We extended **Appendix B.1 (Table 6)** with an ablation study comparing alignment at blocks 6, 8, 12, and 18. The results show that aligning intermediate DiT layers yields the best trade-off between structure preservation and texture generation, while aligning later layers leads to performance degradation.
*   **Lack of human evaluation:** We added user study results in **Appendix D (Table 8)** to assess perceptual quality and validate our results against human judgments.
*   **Lower performance on lower-body items:** We clarified in **Sec. 4.1** that the reduced performance on lower-body garments is primarily due to dataset class imbalance. This limitation is also explicitly noted in **Sec. E of the Appendix**.

---

We appreciate the opportunity to strengthen the work through the reviewers’ feedback. All revisions have been incorporated into the current version of the paper (highlighted in blue). We once again thank the reviewers and the AC for their careful evaluation.


Best regards,

The Authors (Submission #17076)

---

### Meta-Review · Area_Chair_nU7u · 2026-01-07

**Summary:**

This paper proposes TEMU-VTOFF, a diffusion-based framework for virtual try-off, addressing the underexplored task of reconstructing standardized product images from clothed person photos.

Reviewers consistently recognized the novelty and practical relevance of the task, as well as the strength of the dual-DiT architecture, multimodal hybrid attention, and garment alignment module.

The main concerns involved potential caption leakage, clarity of implementation details, metric justification, and generalization.

Based on the rebuttal and revisions, these concerns were largely resolved. I recommend acceptance as a poster.

**Reviewer Concerns:**

Addressed concerns:
- Caption leakage and inference protocol (mTyd): Clearly clarified; captions are generated from person images at inference, with detailed documentation added.
- Implementation clarity (mTyd): Latent-space notation, masked image encoding, and inference-time cost were clarified.
- Evaluation and metrics (Dtrc): PSNR was added consistently, metric choices were justified, and LPIPS limitations discussed.
- Generalization and robustness (Dtrc): Cross-dataset evaluations and robustness analyses were added.
- Human validation (Dtrc, nzHz): A user study was included, aligning results with human preference.
- Ablations and backbones (mTyd, nzHz): Additional ablations (e.g., SigLIP-2, alignment depth) strengthened empirical support.
- Dataset scope and robustness (izfa): Clarifications on face handling, cropped inputs, and added related work improved completeness.

Remaining concerns:

- Performance differences across garment categories (nzHz) persist but are well-explained by dataset imbalance and acknowledged as limitations.

**Reviewer Scores:**

Reviewer mTyd: Score unchanged, already supportive of acceptance.

Reviewer izfa: Score unchanged, already supportive of acceptance.

Reviewer Dtrc: Likely slight increase score given added metrics, generalization tests, and user study.

Reviewer nzHz: Score unchanged reject (4)

---

### Decision · Program_Chairs · 2026-01-26

Accept (Poster)